# Transfer Q⋆: Principled Decoding for LLM Alignment

**Souradip Chakraborty**[1][*]  **Soumya Suvra Ghosal**[1][*]  **Ming Yin**[2]  **Dinesh Manocha**[1]
**Mengdi Wang**[2]  **Amrit Singh Bedi**[3][†]  **Furong Huang**[1][†]

[1]University of Maryland-College Park; [2] Princeton University;
[3] University of Central Florida

## Abstract

Aligning foundation models is essential for their safe and trustworthy deployment. However, traditional fine-tuning methods are computationally intensive and require updating billions of model parameters. A promising alternative, alignment via decoding, adjusts the response distribution directly without model updates to maximize a target reward $r$, thus providing a lightweight and adaptable framework for alignment. However, principled decoding methods rely on oracle access to an optimal Q-function ($Q^*$), *which is often unavailable in practice*. Hence, prior SoTA methods either approximate this $Q^*$ using $Q^{\pi_{\text{sft}}}$ (derived from the reference SFT model) or rely on short-term rewards, resulting in sub-optimal decoding performance. In this work, we propose `Transfer Q⋆`, which implicitly estimates the optimal value function for a target reward $r$ through a baseline model $\rho_{\text{BL}}$ aligned with a baseline reward $r_{\text{BL}}$ (which can be different from the target reward $r$). Theoretical analyses of `Transfer Q⋆` provide a rigorous characterization of its optimality, deriving an upper bound on the sub-optimality gap and identifying a hyperparameter to control the deviation from the pre-trained reference SFT model based on user needs. Our approach significantly reduces the sub-optimality gap observed in prior SoTA methods and demonstrates superior empirical performance across key metrics such as coherence, diversity, and quality in extensive tests on several synthetic and real datasets. The code is available at https://github.com/umd-huang-lab/Transfer-Q.

## 1 Introduction

As artificial intelligence (AI) systems continue to demonstrate super-human performance across various tasks, it is becoming increasingly critical to ensure that such AI systems align well with human preferences, goals, and ethical standards. Alignment via fine-tuning with reinforcement learning from human feedback (RLHF) [38, 34, 9], has proven highly effective [36, 44]. However, aligning LLMs by fine-tuning the model parameters requires gradient updates across several billion parameters (size of LLMs) which require vast computational resources [31, 25] and have a significant environmental impact [15]. Additionally, many SoTA models are not fully open-sourced [44, 36], offering limited access only to certain components like logits, making fine-tuning impossible.

As an alternative to fine-tuning, decoding for alignment has recently emerged as a potential solution [33, 26]. Decoding aims to alter the LLM's response distribution at the token level to align with target reward $r$ without updating the parameters of the LLM. Decoding facilitates alignment by accessing the token-level optimal value function, $Q^*$, which corresponds to the target reward $r$. Tokens are then sampled based on $Q^*$. This approach ensures rapid, efficient, and cost-effective alignment. A detailed discussion of related work is provided in Appendix E.

---

[*]denotes equal contribution

[†]denotes equal advising

38th Conference on Neural Information Processing Systems (NeurIPS 2024).

**A fundamental challenge.** Decoding effectively hinges on accessing an oracle to the optimal value function, $Q^*$, or a token-level optimal policy, which are typically not available in practical scenarios. To address this fundamental challenge, recent studies [33] have adopted a proxy, $Q^{\pi_{\text{sft}}}$, for $Q^*$. However, this approach results in suboptimal decoding due to a distribution shift inherent in approximating the true, unknown Q-function, $Q^*$, as illustrated in Figure 1. This raises a critical question: *Is it possible to devise a more efficient and accurate estimate of the optimal value function $Q^*$ for decoding purposes?* In this work, we affirmatively address this query.

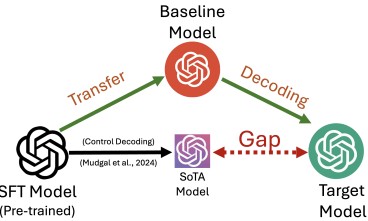
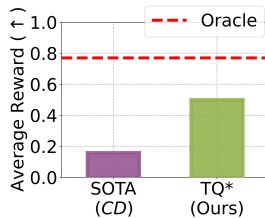

Figure 1: **Left.** This figure highlights the conceptual idea of proposed *transfer decoding* in this work. It clearly shows that the current SoTA method [33] exhibits suboptimality with respect to alignment with the target reward denoted by the dotted red arrow. On the other hand, the proposed transfer decoding method utilizes an immediately available aligned language model called the baseline, which is aligned with some baseline reward $r_{\text{BL}}$ to bridge the gap between the SoTA method and the target model. **Right.** This figure provides empirical evidence of the performance gap of the current SoTA decoding strategy [33] with respect to Oracle (best of $N$ sampling). Our proposed `Transfer Q*` (TQ⋆) reduces the gap and provides a new decoding method.

**Our approach** leverages a crucial observation regarding the key challenges identified: effective decoding requires access to a language model already aligned with the target reward $r$ for trajectory-level response generation. Notably, recent advances in DPO-based methods [38, 3, 32] have facilitated the development of fine-tuned language models—referred to as baseline models—that are capable of generating aligned trajectories [27]. **Our first key idea** involves utilizing these publicly available trajectory-level models to estimate $Q^*$ and subsequently derive the optimal token-level language model for decoding. We term this approach *direct transfer* decoding.

Moreover, it is possible that these publicly available baseline models are aligned with a different baseline reward $r_{\text{BL}}$, rather than the intended target reward $r$. **Our second key idea** addresses this challenge by proposing a novel *indirect transfer* decoding method. We introduce our proposed technique as `Transfer Q*` (TQ⋆), which facilitates efficient and on-the-fly alignment through decoding. We note that `Transfer Q*` proves effective even when there are substantial discrepancies between the target and baseline rewards.

We summarize our **major contributions** as follows.
**(1) A novel concept of transfer decoding (TQ⋆).** We introduce a novel concept of *transfer decoding* in a principled manner by leveraging already-available baseline language models aligned either with the target reward (direct transfer) or with some other (significantly) different baseline reward (indirect transfer). Our proposed approach TQ⋆ efficiently reduces the sub-optimality gap inherent in previous SoTA decoding methods, as highlighted in Figure 1.
**(2) Theoretical characterization of TQ⋆.** We provide a rigorous theoretical characterization of the optimality of `Transfer Q*`. Specifically, we derive an upper bound on the gap between the optimal LLM policy and the LLM decoding policy resulting from TQ⋆ (see Theorem 1). Additionally, we present a principled approach to control the deviation of the resulting policy from the pre-trained SFT language model $\pi_{\text{sft}}$ (see Theorem 1, statement 2). This identifies hyperparameters that allow users to specify the desired amount of deviation or improvement based on their specific needs.
**(3) Experimental evaluations.** We provide a detailed empirical evaluation of TQ⋆ in various decoding tasks in both real and simulated settings, comparing against SoTA baselines, including DPO [38], ARGS [26], and CD [33]. Empirical results demonstrate consistent superiority over the baselines. Notably, TQ⋆ surpasses the current SoTA decoding strategy CD [33], achieving an improvement of up to 1.45x in average reward and $67.34\%$ in GPT-4 based win-tie rate. We further evaluate and compare several attributes of the text generated by our algorithm (such as coherence, diversity, and quality) against baselines, demonstrating the superiority of our algorithm in all these aspects.

# 2 Problem Formulation: Alignment via Controlled Decoding

## 2.1 Token-level Markov Decision Process

Since the control decoding procedure operates at the token level, before formulating the problem mathematically, we start by defining a token-level Markov decision process (MDP) in the context of LLMs. Let us consider a token-level MDP $\mathcal{M} := \{\mathcal{S}, \mathcal{A}, P, R\}$ with the state-space $\mathcal{S}$ consisting of the concatenated sequence of tokens and the action space $\mathcal{A}$ representing the space of the next token which is essentially the vocabulary $\mathcal{V}$.

**Next-token generator: the token-level policy $\pi$.** Given a state $\mathbf{s}_t = [\mathbf{x}, \mathbf{y}_{<t}] \in \mathcal{S}$, which is a sequence of tokens containing the prompt/query $\mathbf{x} := \{x_1, x_2, \cdots, x_N\}$ appended with the $t$ tokens $\mathbf{y}_{<t} := \{y_0, y_1, \cdots, y_{t-1}\}$ generated so far, an LLM is a token-level decoding policy $\pi$ that generates the action (i.e., the next token) $a_t = y_t$ via sampling $y_t \sim \pi(\cdot \mid \mathbf{s}_t)$. The transition $P$ to the next state $\mathbf{s}_{t+1}$ is deterministic: $\mathbf{s}_{t+1} = [\mathbf{x}, \mathbf{y}_{<t}, y_t]$, the concatenation of the current state and action.

**Response generator: the trajectory-level policy $\rho$.** We denote the trajectory level probability by $\rho_\pi(\mathbf{y}|\mathbf{x}) = \prod_{t=1}^{T} \pi(y_t|[\mathbf{x}, \mathbf{y}_{<t}])$. Some commonly used sampling techniques in the literature include Beam Search [17], Top-p sampling [20], and Top-k Sampling [16].

**From trajectory-level reward $r$ to token-level reward $R$.** Successful decoding depends on sampling from a token-level policy that yields high rewards, reflecting the inherently token-level nature of the decoding process. However, as detailed in Appendix F, we obtain a reward model $r(\mathbf{x}, \mathbf{y})$, only at the trajectory-level rather than the desired token-level, by fitting feedback on human preferences. To close the gap, similar to existing literature [33], we define the token-level reward $R(\mathbf{x}, y_t)$ from the trajectory-level reward model $r(\mathbf{x}, \mathbf{y})$ as follows:

$$R(\mathbf{x}, y_t) := \begin{cases} 0, & y_t \neq \text{EOS} \\ r(\mathbf{x}, \mathbf{y}_{<t}), & y_t = \text{EOS}, \end{cases} \tag{1}$$

where $\text{EOS} \in \mathcal{V}$ represents the end of sequence token. The token-level reward in (1) implies that we only receive a reward once we have the full sequence/response, otherwise, no reward.

**Action-value function $Q^\pi$ for $\pi$ and optimal $Q^*$ for optimal $\pi^*$.** From the token-level reward $R(\mathbf{x}, y_t)$, we can define the action-value function associated with the reward as

$$Q^\pi(\mathbf{s}_t, a_t) = Q^\pi([\mathbf{x}, \mathbf{y}_{<t}], y_t) = \mathbb{E}\left[\sum_i R([\mathbf{x}, \mathbf{y}_{<t}], z_i) \mid z_0 = y_t, z_i \sim \pi(\cdot|\mathbf{s}_{t+i})\right], \tag{2}$$

where $\mathbf{s}_{t+i} := [\mathbf{s}_t, z_0, z_1, \cdots, z_{i-1}]$ and expectation is over the randomness due to the sampling from token level language model $\pi$. The optimal Q-function from the definition in equation (2) is given by

$$Q^*(\mathbf{s}_t, a_t) = \max_\pi Q^\pi(\mathbf{s}_t, a_t). \tag{3}$$

The optimization problem in (3) denotes an unconstrained objective as in standard reinforcement learning; however, in the context of alignment for LLMs, we also need to consider the distance of optimal aligned policy to the pre-trained unaligned token-level policy $\pi_{\text{sft}}$ [37, 41, 38].

## 2.2 Principled Decoding for LLM Alignment

In this section, we will formulate the problem of aligning LLMs during deployment via a controlled decoding procedure as initially discussed in [33, 26].

**Decoding process.** We start by defining what decoding means in the context of LLMs. We consider access to a pre-trained unaligned language model $\pi_{\text{sft}}$ which takes in prompt $\mathbf{x}$ as an input and generates a response $\mathbf{y} = [y_0, y_1, \cdots, \text{EOS}]$ token by token by sampling $y_t \sim \pi_{\text{sft}}(\cdot|[\mathbf{x}, \mathbf{y}_{\leq t}])$ for all $t$. This token-by-token generation of response is called decoding in LLMs. Hence, the natural next question arises if we can control the decoding process and generate responses that are aligned with respect to a target reward function $r(\mathbf{x}, \mathbf{y})$. The quest to answer this question has given rise to an interesting research problem of utilizing decoding for LLM alignment [33].

**LLM alignment via decoding.** The problem of LLM alignment via decoding can be formally defined as solving for the optimal decoding policy $\pi_{\text{dec}}^*$ under the token level MDP $\mathcal{M}$ as

$$\pi_{\text{dec}}^*(\cdot|\mathbf{s}_t) := \arg\max_{\pi \in \Pi} \mathbb{E}_{z \sim \pi(\cdot|\mathbf{s}_t)} \left[Q^*(\mathbf{s}_t, z)\right] - \alpha \mathbb{D}_{\text{KL}}\left[\pi(\cdot|\mathbf{s}_t)||\pi_{\text{sft}}(\cdot|\mathbf{s}_t)\right], \tag{4}$$

where $\mathbf{s}_t = [\mathbf{x}, \mathbf{y}_{<t}]$ and $Q^*(\mathbf{s}_t, z)$ denotes the optimal state-action value function for the token-level MDP $\mathcal{M}$ defined in (3). We remark that the KL regularization in equation (4) ensures that the optimal decoding policy $\pi^*_{\mathsf{dec}}$ remains in the close neighborhood of the pre-trained model $\pi_{\mathsf{sft}}$ which contains other important properties. In (4), $\alpha > 0$ denotes the alignment hyperparameter which controls the trade-off between the objective of maximizing the target reward $r$ return and the closeness to $\pi_{\mathsf{sft}}$. We can write the closed-form solution of the problem in (4) as

$$\pi^*_{\mathsf{dec}}(z|\mathbf{s}_t) = \pi_{\mathsf{sft}}(z|\mathbf{s}_t)\frac{\exp\left(\frac{1}{\alpha}Q^*(\mathbf{s}_t, z)\right)}{C_\alpha}, \tag{5}$$

where $C_\alpha := \sum_z \pi_{\mathsf{sft}}(z|\mathbf{s}_t)\exp\left(\alpha Q^*(\mathbf{s}_t, z)\right)$ is the normalizing constant for state $\mathbf{s}_t$. Although the close form expression in (5) poses an interesting form, it is difficult to implement it in practice due to the various challenges we will discuss next.

**Challenges of implementing (5): oracle access to the optimal $Q^*$.** A major challenge in implementing the aligned decoding policy in (5), lies in the requirement of access to the optimal $Q^*(\mathbf{s}_t, z)$ in (5) for each state-action pair $(\mathbf{s}_t, z)$, which is unavailable in practice. To emphasize that, first we note that $Q^*(\mathbf{s}_t, z)$ in (3) can be written using the trajectory level reward $r$ as

$$Q^*(\mathbf{s}_t, z) = \mathbb{E}_{\tau \sim \rho^*(\cdot|\mathbf{s}_t, z)}\left[r([\mathbf{x}, \mathbf{y}_{<t}, z], \tau)\right], \tag{6}$$

where $\tau$ denotes the trajectory $\tau := \{z_1, z_2, \cdots, z_T\}$, and $\rho^*(\cdot|\mathbf{s}_t, z) = \prod_{i=1}^T \pi^*(z_i|[\mathbf{s}_{t+i}])$ represents the distribution over the trajectory level response induced by the optimal policy $\pi^*(\cdot|\mathbf{s}_t)$ (cf. (3)). From (6), we note that the optimal $Q^*(\mathbf{s}_t, z)$ relies on the trajectory/response generated by the optimal $\rho^*$ which is unknown. This creates a bottleneck in efficiently deploying the decoding policy in (5) for alignment with target reward $r$. Next, we discuss how some existing approaches in the literature deal with this fundamental issue and what the limitations are.

**Limitations in existing approaches.** An interesting approach called *controlled decoding (CD)* is proposed in recent literature by Mudgal et al. [33], and it approximates $Q^*(\mathbf{s}_t, z)$ by $Q^{\pi_{\mathsf{sft}}}(\mathbf{s}_t, z)$, which is tractable and easily computable due to availability of $\pi_{\mathsf{sft}}$. However, this approximation results in significant suboptimal performance, as described in Figure 1 (right). Given this limitation of existing approaches and the above-mentioned fundamental challenge of decoding, we pose the question: Is it possible to provide a better estimate of $Q^*$ for decoding? We answer this affirmatively in the following section by introducing a novel concept of *transfer decoding*.

## 3 Proposed Approaches: Alignment via `Transfer Q`$^\star$

**Our key ideas of transfer decoding.** Our proposed approach hinges on an interesting observation that recent advancements in alignment, particularly through direct preference optimization (DPO)-based approaches [38], have led to the development of open source freely available fine-tuned language models [27, 53]. We call such aligned models as baseline models, that generate trajectory responses in an aligned manner. ***Our first key idea*** is to utilize these baseline models aligned with target reward $r$, if available, to estimate $Q^*$ and subsequently derive the optimal token-level language model for decoding. We remark that baseline models are aligned at the trajectory level (see Appendix F for details), while decoding requires the optimal models at the token level. We term this approach *direct transfer decoding*. However, it is possible that the available aligned baseline model is aligned with a different baseline reward $r_{\mathsf{BL}}$ instead of the target reward $r$. Hence, ***our second key idea*** addresses this issue by proposing a novel method called *indirect transfer decoding*. We discuss both the ideas and detailed algorithms in detail next.

### 3.1 Direct Transfer Decoding

For the direct transfer, we start by considering that we are given a target reward model $r$ and an unaligned pre-trained SFT language model given by $\pi_{\mathsf{sft}}$; the corresponding trajectory-level response distribution is given by $\rho_{\mathsf{sft}}$. We are operating under the assumption that we have a baseline model $\rho_{\mathsf{BL}}(\mathbf{y}|\mathbf{x})$ with target reward $r$. We note that the closed-form expression for the RLHF-aligned optimal model $\rho_{\mathsf{BL}}$ can be written as (see Appendix F, Equation (25)) follows:

$$\rho_{\mathsf{BL}}(\mathbf{y}|\mathbf{x}) = \frac{1}{Z_r(\mathbf{x})}\rho_{\mathsf{sft}}(\mathbf{y}|\mathbf{x})\exp\left(\frac{1}{\beta}r(\mathbf{x}, \mathbf{y})\right), \tag{7}$$

where $Z_r(\mathbf{x})$ is the normalizing constant and $\beta > 0$ is the trajectory-level alignment parameter. We note that the trajectory level optimal policy in (7) is usually obtained in the literature via the fine-tuning stage and efficient algorithms such as DPO [38].

**Estimating $Q^*$ for direct transfer.** Now we get back to the fundamental bottleneck of optimal decoding, which lies in estimating the token-level optimal $Q^*(\mathbf{s}_t, z)$. We propose providing a solution to the problem with the help of $\rho_{\text{BL}}(\mathbf{y}|\mathbf{x})$. As defined in (6), we begin by considering

$$Q^*(\mathbf{s}_t, z) = \mathbb{E}_{\tau \sim \rho^*(\cdot|\mathbf{s}_t, z)} \left[ r([\mathbf{s}_t, z], \tau) \right], \tag{8}$$

where $\rho^*(\cdot|\mathbf{s}_t, z) := \arg\max_\rho \mathbb{E}_{\tau \sim \rho(\cdot|\mathbf{s}_t, z)} \left[ r(\mathbf{s}_t, \tau) \right]$. However, we know that the available baseline model $\rho_{\text{BL}}(\mathbf{y}|\mathbf{x})$ in (7) is the solution of following optimization problem:

$$\rho_{\text{BL}}(\cdot|\mathbf{x}) := \arg\max_\rho \mathbb{E}_{\tau \sim \rho(\cdot|\mathbf{x})} \left[ r(\mathbf{x}, \tau) \right] - \beta \mathbb{D}_{\text{KL}} \left[ \rho(\cdot|\mathbf{x}) \,||\, \rho_{\text{sft}}(\cdot|\mathbf{x}) \right], \tag{9}$$

which constraints the drift of the optimal distribution $\rho_{\text{BL}}$ from the trajectory level reference policy $\rho_{\text{sft}}$ with the Kl divergence term. We propose approximating the optimal $Q^*(\mathbf{s}_t, z)$ for decoding by

$$\text{TQ}^\star(\mathbf{s}_t, z) = \mathbb{E}_{\tau \sim \rho_{\text{BL}}(\cdot|\mathbf{s}_t, z)} \left[ r([\mathbf{s}_t, z], \tau) \right]. \tag{10}$$

With the definition in (10), we propose obtaining our token-level decoding policy $\pi^*_{\text{Alg}}(\cdot|\mathbf{s}_t)$ for the token-level MDP as

$$\pi^*_{\text{Alg}}(\cdot|\mathbf{s}_t) := \arg\max_{\pi \in \Pi} \mathbb{E}_{z \sim \pi(\cdot|\mathbf{s}_t)} \left[ \text{TQ}^\star(\mathbf{s}_t, z) \right] - \alpha \mathbb{D}_{\text{KL}} \left[ \pi(\cdot|\mathbf{s}_t) || \pi_{\text{BL}}(\cdot|\mathbf{s}_t) \right], \tag{11}$$

where $\pi_{\text{BL}}(\cdot|\mathbf{s}_t)$ is the token-level language model, which induces the trajectory level model $\rho_{\text{BL}}$. Due to the strong convexity of the objective in (11) owing to the additional KL regularization term, we get the closed-form solution of $\pi^*_{\text{Alg}}(\cdot|\mathbf{s}_t)$ as

$$\pi^*_{\text{Alg}}(z|\mathbf{s}_t) = \frac{1}{\widetilde{C}_\alpha(\mathbf{s}_t)} \pi_{\text{BL}}(z|\mathbf{s}_t) \exp\left( \frac{1}{\alpha} \cdot \text{TQ}^\star(\mathbf{s}_t, z) \right), \tag{12}$$

where $\widetilde{C}_\alpha(\mathbf{s}_t)$ is the normalizing constant. We summarize the proposed approach in Algorithm 1.

## 3.2 Indirect Transfer Decoding

We remark that, for the direct transfer decoding, we started with the assumption that the available baseline model $\rho_{\text{BL}}$ is aligned with the target model only. In practice, however, it is possible that the freely available baseline model $\rho_{\text{BL}}$ is actually aligned with some other reward function we call baseline $r_{\text{BL}}$ rather than the target reward $r$. We argue that this condition is even more easily satisfied under the ongoing active research scenario in alignment because we have easy access to open-source, well-aligned LLMs trained on various reward functions [27, 52, 13, 38]. Under this setting of baseline reward $r_{\text{BL}}$ and language model $\rho_{\text{BL}}$, we define our novel indirect transfer decoding process as follows.

**The Transfer Process.** The baseline language model $\rho_{\text{BL}}$ is also an RLHF aligned model corresponding to reward function $r_{\text{BL}}$. It holds that:

$$r_{\text{BL}}(\mathbf{x}, \mathbf{y}) = \beta \log \frac{\rho_{\text{BL}}(\mathbf{y}|\mathbf{x})}{\rho_{\text{sft}}(\mathbf{y}|\mathbf{x})} + \beta \log Z_{\text{BL}}(\mathbf{x}). \tag{13}$$

where $Z_{\text{BL}}(\mathbf{x})$ is the corresponding partition function. From the closed-form expression in (7), it holds for the target reward $r$ that

$$r(\mathbf{x}, \mathbf{y}) = \beta \log \frac{\rho_r(\mathbf{y}|\mathbf{x})}{\rho_{\text{sft}}(\mathbf{y}|\mathbf{x})} + \beta \log Z_r(\mathbf{x}). \tag{14}$$

Using equations (14) and (13), we can obtain the trajectory-level optimal policy $\rho_r(\mathbf{y}|\mathbf{x})$ for the target reward $r(\mathbf{x}, \mathbf{y})$ as :

$$\rho_r(\mathbf{y}|\mathbf{x}) = \underbrace{\rho_{\text{BL}}(\mathbf{y}|\mathbf{x}) \exp\left[ \frac{1}{\beta}(r(\mathbf{x}, \mathbf{y}) - r_{\text{BL}}(\mathbf{x}, \mathbf{y})) \right]}_{:=\widetilde{\rho}_r(\mathbf{y}|\mathbf{x})} \frac{Z_{\text{BL}}(\mathbf{x})}{Z_r(\mathbf{x})}, \tag{15}$$

---

**Algorithm 1** `Transfer Q*`: LLM Alignment via Transfer Decoding

---

1: **Input:** Trajectory level baseline model $\rho_{\text{BL}}(\mathbf{y}|\mathbf{x})$ aligned with baseline reward $r_{\text{BL}}$, target reward $r$, token-level baseline policy $\pi_{\text{BL}}$, number of tokens sampled $k$, decoding alignment parameter $\alpha$, vocabulary set $\mathcal{V}$.

2: **for** $t = 0, \ldots, T$ **do**

3:     Current state : $\mathbf{s}_t = [\mathbf{x}, \mathbf{y}_{<t}]$, where $\mathbf{x}$ is prompt and $\mathbf{y}_{<t} = [y_0, y_1, \cdots, y_{t-1}]$

4:     Sample top-k tokens using token-level baseline policy $\pi_{\text{BL}}$ and store as $\hat{\mathcal{V}} = \{z_i : z_i \sim \pi_{\text{BL}}(\cdot|\mathbf{s}_t)\}_{i=1}^k$

5:     **for** $z \in \hat{\mathcal{V}}$ **do**

6:         **if** $r_{\text{BL}} = r$ **then** (Direct transfer)

7:             **Evaluate** $\texttt{TQ}^\star(\mathbf{s}_t, z) = r([\mathbf{s}_t, z], \tau)$, where $\tau \sim \rho_{\text{BL}}(\cdot|[\mathbf{s}_t, z])$

8:         **else**    (Indirect transfer)

9:             **Evaluate** $\texttt{TQ}^\star(\mathbf{s}_t, z) = w \cdot r([\mathbf{s}_t, z], \tau)$ where $\tau \sim \rho_{\text{BL}}(\cdot|[\mathbf{s}_t, z])$, $w = \frac{\rho_r(\tau|[\mathbf{s}_t, z])}{\rho_{\text{BL}}(\tau|[\mathbf{s}_t, z]))}$

10:        **Compute** decoding score for token $z$: $g_z = \frac{1}{\alpha}\texttt{TQ}^\star(\mathbf{s}_t, z) + \log \pi_{\text{BL}}(z|\mathbf{s}_t)$

11:    **Next token** $y_t = \arg\max_{z \in \hat{\mathcal{V}}} g_z$

12:    **Next state** $\mathbf{s}_{t+1} \leftarrow [\mathbf{s}_t, y_t]$

13: Return $\mathbf{y} = [y_0, \ldots, y_T]$

---

where we note that $\widetilde{\rho}_r(\mathbf{y}|\mathbf{x})$ is the unnormalized probability with the normalization factory $\tilde{Z}(\mathbf{x}) := \frac{Z_r(\mathbf{x})}{Z_{\text{BL}}(\mathbf{x})}$. We show in the Appendix H that $\tilde{Z}(\mathbf{x})$ is the normalization constant for $\widetilde{\rho}_r(\mathbf{y}|\mathbf{x})$. We emphasize that calculating the trajectory-level optimal language model for the target reward function $r$ in (15) is the crucial step in estimating the optimal $Q^*$ for the token-level MDP $\mathcal{M}$ for our decoding.

**Estimating $Q^*$ for indirect transfer.** Similar to the direct transfer setting, we propose approximating the optimal $Q^*(\mathbf{s}_t, z)$ for decoding by using $\rho_r(\mathbf{y}|\mathbf{x})$ in (15) as

$$\texttt{TQ}^\star(\mathbf{s}_t, z) = \mathbb{E}_{\tau \sim \rho_r(\cdot|\mathbf{s}_t, z)}\left[r([\mathbf{s}_t, z], \tau)\right]. \tag{16}$$
$$= \mathbb{E}_{\tau \sim \rho_{\text{BL}}(\cdot|\mathbf{s}_t, z)}\left[\frac{\rho_r(\mathbf{y}|\mathbf{x})}{\rho_{\text{BL}}(\mathbf{y}|\mathbf{x})}r([\mathbf{s}_t, z], \tau)\right]$$

where we use the importance sampling trick and then utilizing equation (15), we can get the $Q^*$ for indirect transfer. Now, following (11) and (12), we can write

$$\pi^*_{\texttt{Alg}}(\cdot|\mathbf{s}_t) := \arg\max_{\pi \in \Pi} \mathbb{E}_{z \sim \pi(\cdot|\mathbf{s}_t)}\left[\texttt{TQ}^\star(\mathbf{s}_t, z)\right] - \alpha \mathbb{D}_{\text{KL}}\left[\pi(\cdot|\mathbf{s}_t)||\pi_r(\cdot|\mathbf{s}_t)\right], \tag{17}$$

where $\pi_r(\cdot|\mathbf{s}_t)$ is the token level probability derived from the trajectory level policy $\rho_r(\cdot|\mathbf{s}_t)$ in (15). Following (11) and (12), we can obtain our optimal decoding policy for the indirect transfer as well. We summarize the proposed approach in Algorithm 1.

### 3.3 Theoretical Results and Insights

This subsection provides the theoretical analysis of our `Transfer Q*` algorithm under direct transfer setup. Existing works [33, 18] leverage the reward-KL tradeoff curve to measure the *reward gain* versus *deviation from reference policy*. Ideally, a good algorithm should achieve high rewards while staying close to the reference policy (be KL-efficient). We follow these two aspects and consider two performance metrics: (1) suboptimality gap and (2) KL divergence between our algorithm's policy and the SFT policy. Specifically, we borrow the notion of a suboptimality-gap from offline RL literature [1] and define it in terms of value function difference for any prompt $\mathbf{x}$ as

$$\texttt{Sub-Gap}(\mathbf{x}) := V^*(\mathbf{x}) - V_{\texttt{Alg}}(\mathbf{x}). \tag{18}$$

where $V^*(\mathbf{x}) = \max_\rho \mathbb{E}_{\tau \sim \rho(\cdot|\mathbf{x})}[r(\mathbf{x}, \tau)]$, $V_{\texttt{Alg}}(\mathbf{x}) = \mathbb{E}_{\tau \sim \rho^*_{\texttt{Alg}}(\cdot|\mathbf{x})}[r(\mathbf{x}, \tau)]$, and $\rho^*_{\texttt{Alg}}$ represents the distribution over the trajectories induced by the token level policy $\pi^*_{\texttt{Alg}}(\cdot|\mathbf{x})$ in (17). The KL divergence between our algorithm and the reference policy is denoted by $\mathbb{D}_{\text{KL}}\left(\rho^*_{\texttt{Alg}}(\cdot|\mathbf{x}), \rho_{\texttt{sft}}(\cdot|\mathbf{x})\right)$. We present our main theorem as follows, and the full proof is deferred to Appendix G.

**Theorem 1.** *For the proposed `Transfer Q*` Algorithm 1, the following results hold.*

*(1) Suboptimality gap for all $\mathbf{x}$ is upper bounded as*

$$\texttt{Sub-Gap}(\mathbf{x}) \leq \beta \mathbb{D}_{KL}\big(\rho^*(\cdot \,|\, \mathbf{x}), \rho_{sft}(\cdot \,|\, \mathbf{x})\big) - \alpha h_\alpha(\mathbf{x}), \tag{19}$$

*where $\beta$ is defined in (9) for baseline policy, and $\alpha$ is defined in (17) for decoding process. Here $h_\alpha(\mathbf{x}) \geq 0$ and its formula is defined in Appendix G.*

*(2) Assume reward satisfies $0 \leq r \leq r_{max}$.then the divergence to SFT policy is given by*

$$\mathbb{D}_{KL}\big(\rho^*_{Alg}(\cdot \,|\, \mathbf{x}), \rho_{sft}(\cdot \,|\, \mathbf{x})\big) \leq \left(\frac{1}{\beta} + \frac{1}{\alpha}T\right) r_{max}. \tag{20}$$

**Remark 1 ("Double Robustness" of `Transfer Q*`).** Theorem 1 indicates the suboptimality is bounded by $\beta \mathbb{D}_{KL}(\rho^*, \rho_{sft})$, and this guarantees our algorithm will achieve high accuracy in two cases. 1. The penalty parameter $\beta$ is small, 2. The SFT policy $\rho_{sft}$ is close to $\rho^\star$. In addition, our decoding design (17) is also effective for improving the performance. This is due to $-\alpha h_\alpha \leq 0$. Our decoding coefficient $\alpha$ could further reduce the suboptimality gap when it is appropriately tuned (*e.g.* choose $\alpha^* = \arg\min_\alpha \alpha h_\alpha$).

**Remark 2 (KL-Efficiency of `Transfer Q*`).** Via Theorem 1, the deviation from our algorithm to the reference policy is jointly controlled by parameter $\beta$ and decoding parameter $\alpha$. When both parameters are set large, our algorithm is more conservative and hence more KL-efficient. When the parameters are small, the KL deviation becomes larger.

## 4 Experimental Evaluations

We present a comprehensive empirical analysis of both direct and indirect `Transfer Q*`, tested across various open-source datasets and state-of-the-art models [27]. Our findings demonstrate `TQ*`'s effectiveness in aligning language model outputs with specific target rewards. For implementation, we set the number of tokens sampled $k = 10$ and the decoding alignment parameter $\alpha = 1$. We report ablations in Appendix J.3. Reproducibility is ensured through the use of publicly available resources.

Table 1: Summary of the datasets and model architectures used for experimental evaluations in Section 4.1.

| Dataset | | Model Architectures | | | Reward Preference |
|---|---|---|---|---|---|
| | | SFT | DPO | Reward | |
| Evaluation-1 | UltraFeedback [12] | Mistral-7B-$\alpha$ [24] | Zephyr-7B-$\alpha$ [46] | Mistral-7B-$\alpha$ [24] | Relevant, Helpful, and Ethical responses. |
| Evaluation-2 | HH-RLHF [5] | Pythia-6.9B [7] | Pythia-6.9B [7] | Pythia-6.9B [7] | Helpful and Harmless responses. |
| Evaluation-3 | Berkeley Nectar [53] | OpenChat 3.5-7B [47] | Starling-7B-$\alpha$ [53] | Mistral-7B-$\alpha$ [24] | Accurate, Helpful, and Harmless responses. |
| Evaluation-4 | UltraFeedback [12] | Llama-2-7B [45] | Tulu-v2-7B [23] | Mistral-7B-$\alpha$ [24] | Relevant, Helpful, and Ethical responses. |
| Evaluation-5 | UltraFeedback [12] | Mistral-7B-$\alpha$ [24] | Zephyr-7B-$\alpha$ [46] | Gemma-7B [6] | Relevant, Helpful, and Ethical responses. |
| Evaluation-6 | UltraFeedback [12] | Mistral-7B-$\alpha$ [24] | Zephyr-7B-$\alpha$ [46] | Gemma-2B [6] | Relevant, Helpful, and Ethical responses. |

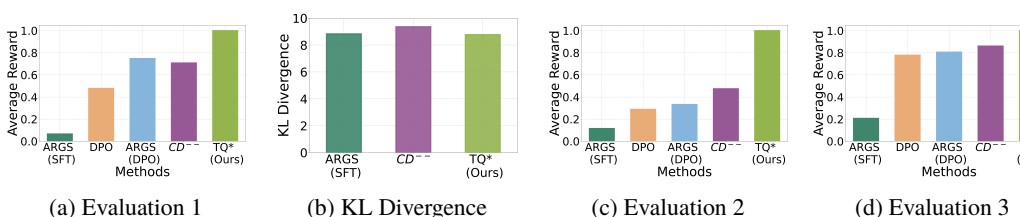

|        (a) Evaluation 1        |        (b) KL Divergence        |        (c) Evaluation 2        |        (d) Evaluation 3        |

Figure 2: In plots (a), (c), and (d) we present the normalized average reward values obtained using the corresponding setup outlined in Table 1. ARGS (SFT) and ARGS (DPO) refer to the reward modeling approach described in [26] to the SFT and DPO model respectively. Our analysis reveals that across all setups, `TQ*` consistently outperforms other competitive baselines summarized in Table 1, demonstrating its superior efficacy. We report results on other evaluation setups in Appendix J. In (b), we compare (for Evaluation-1 setup) the trajectory-level KL Divergence between different decoding policies and the base model $\rho_{sft}$ to show the effectiveness of the proposed approach compared to the state-of-the-art.

**Evaluation Methodology.** For evaluation, we compare the performance of the response generated by the language model corresponding to each prompt in the test dataset. Following [26], we limit the maximum length of the prompt and generated continuation to 128 and 2048 tokens, respectively. For all baselines, we utilize a greedy-based sampling method. The quality of the generated responses is assessed based on multiple attributes (including reward achieved, win-tie rate, coherence, diversity,

etc.) using the following evaluation metrics [26]: **Average Reward:** We report the mean of the rewards for generations corresponding to all prompts in the test set. A higher mean reward score signifies that the model's outputs are better aligned with the attributes represented in the reward model. **Diversity:** This metric measures the ability to generate texts with a wide range of vocabulary. Specifically, it calculates the frequency of repeated n-grams in text. **Coherence:** We assess the semantic closeness between each prompt and its generated response using the cosine similarity of their SimCSE-generated [43] embeddings.

## 4.1 `Transfer Q⋆`: Evaluations with Direct Transfer Decoding

**Experiment Details.** For the direct transfer experiments, we consider our baseline model as a DPO [38] fine-tuned model aligned with the target reward. To evaluate the performance of `Transfer Q⋆` (denoted as TQ⋆ in figures), we conduct experiments across multiple datasets and model architectures, encompassing 6 distinct tasks. Our experimentation is primarily based on the Ultrafeedback [12], Berkeley Nectar [53], and HH-RLHF (Helpful and Harmless) [5] datasets. For each task, we utilize the DPO [38] fine-tuned model as an aligned policy, with configurations listed in Table 1. This comprehensive approach allows us to gauge the adaptability and efficacy of TQ⋆ in various contexts, providing a robust measure of its capabilities.

**Results Discussion.** In Figure 2, we present the normalized average rewards for the first three setups detailed in Table 1. We report the results for other setups in Appendix J.1. We compare our proposed method TQ⋆ with competitive existing approaches such as ARGS [26], CD⁻⁻ [33][3], and DPO [38]. To provide a clearer comparison of results, we normalize the average rewards (further details of normalization in Appendix I.1). We observe that across all setups, TQ⋆ consistently outperforms the existing approaches by a large margin, highlighting its efficacy in learning token-level optimal policy during inference. Further, in Figure 3, we report that TQ⋆ not only produces responses with high rewards but also outperforms other decoding strategies in terms of diversity and coherence.

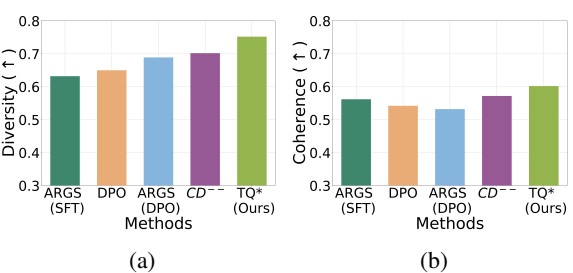

(a)                    (b)

Figure 3: **Diversity and Coherence analysis of generated responses.** We observe that the responses generated using TQ⋆ obtain the highest coherence and diversity. These results are based on the prompts from the Berkeley Nectar dataset.

**GPT-4 Evaluation.** To further understand the quality of the responses generated, we employ a GPT-4-based evaluation framework. Specifically, we use GPT-4 as a surrogate for human assessment. In this method, we prompt GPT-4 to assess and rate two responses on the same prompt on a scale from 1 to 10, focusing on criteria such as relevance, accuracy, and insightfulness. For this, we randomly sample 300 prompts from the test set and compare the response between TQ⋆ and other competitive decoding methods. We present the GPT-4 evaluation results in Table 2, measured by the percentage of win-ties of our method over the

Table 2: **GPT-4 Based Evaluation.** We prompt GPT-4 to rate responses from various decoding strategies on relevance, accuracy, and insightfulness, scoring them from 1 to 10. A higher win-tie percentage indicates our method's effectiveness in generating contextually relevant and accurate responses.

| Ours | vs. | Methods | Win-Tie (%) ↑ | | |
|------|-----|---------|--------------|--------------|--------------|
| | | | **Evaluation-1** | **Evaluation-2** | **Evaluation-3** |
| TQ⋆ | | ARGS-SFT | 86.67 | 72.00 | 75.34 |
| TQ⋆ | | DPO | 70.67 | 77.34 | 70.00 |
| TQ⋆ | | ARGS-DPO | 68.00 | 71.33 | 74.00 |
| TQ⋆ | | CD⁻⁻ | 66.67 | 65.34 | 67.34 |

alternative decoding strategies. A higher percentage indicates that our proposed method is more proficient in generating responses that exhibit better alignment with human preferences. Our analysis in Table 2 shows that TQ⋆ consistently has a higher win-tie percentage compared to other decoding approaches, reaffirming its efficacy.

---

[3]Due to unavailability of code base, we compare using an approximate version of CD [33] in which we do not train an adapter module.

## 4.2 `Transfer Q*`: Evaluations with Indirect Transfer Decoding

**Synthetic experiments.** In the synthetic experiments, we simulate four scenarios to examine shifts in the reward distribution between source and target reward models. These scenarios are instrumental in elucidating the advantages of our proposed method. The shift in reward distribution is achieved through model intervention techniques, such as inducing sparsity in the final linear layer. Our analysis utilizes the UltraFeedback [12] and Berkeley Nectar [53] datasets. The specifics of the models utilized are detailed in Table 4 in Appendix I.2. For each dataset, we design two transfer tasks by modulating the reward distribution. We visualize the shift in reward distribution on the Ultrafeedback dataset in Figure 4 (a) and (c) respectively. Given that this is a synthetic setup, no DPO-aligned policies exist for the newly generated reward distribution, which underscores the significance of addressing the transfer problem. We present the results for this analysis on the Ultrafeedback [12] dataset in Figure 4. Due to space constraints, we report the results on the Berkeley Nectar [53] in Appendix J.2. We make the following key observations: (1) Our proposed decoding framework consistently attains the highest average reward and outperforms other competitive strategies. (2) The variant of our decoding strategy obtained by direct transfer to the target reward, i.e., DT has subpar performance.

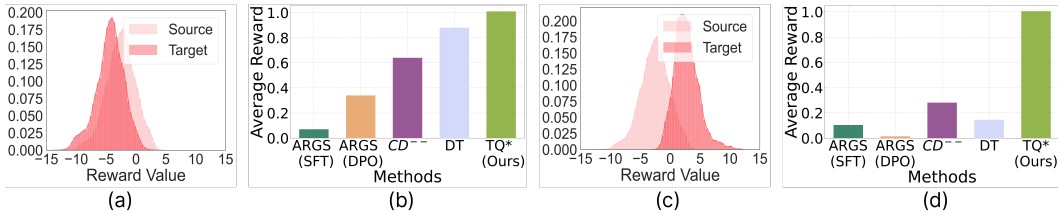

Figure 4: **Evaluation for Synthetic Indirect Transfer Tasks.** We plot the distribution of the reward values for the source and two transfer tasks on the Ultrafeedback in (a) and (c). The reward model architecture is Mistral-7B-$\alpha$ [24]. In (b) and (d), we compare the normalized average reward scores for competitive decoding strategies. We represent the variant of our decoding strategy with direct transfer as DT. We observe that TQ$^\star$ consistently outperforms the other baselines. Results on other datasets are reported in Appendix J.2.

**Real transfer experiments.** To further evaluate our proposed approach on real reward transfer, we consider two transfer setups as outlined in Table 5 in Appendix I.3. We illustrate the distribution shift in reward values on UltraFeedback [12] and HH-RLHF [5] datasets in Figure 5 (a) and (c) respectively. We compare the normalized average reward values of different decoding strategies in Figure 5. Consistent with our findings from the synthetic experiments, our proposed method consistently outperforms other decoding strategies, underscoring its effectiveness in real transfer tasks as well. Notably, we observe that even when there is a significant shift in the source and target reward values, as evident in Figure 5 (c), TQ$^\star$ outperforms all competitive decoding approaches in terms of average reward.

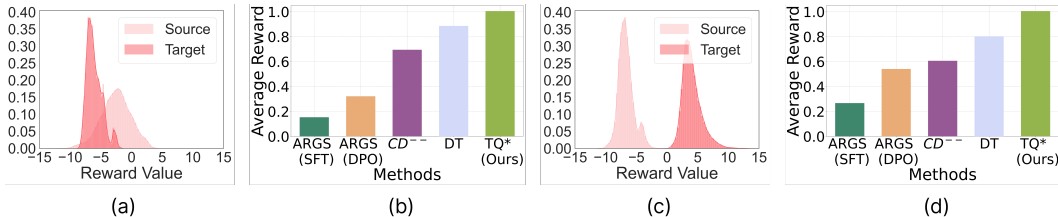

Figure 5: **Evaluation for Real Indirect Transfer Tasks.** In (a) and (c), we visualize the distribution shift in reward values between the source and target for Setup-1 and Setup-2, respectively, as outlined in Table 5. In (b) and (d), we report the normalized average reward scores of different decoding strategies corresponding to Setup-1 and Setup-2, respectively.

## 5 Conclusions

In this paper, we introduce `Transfer Q*`, a novel decoding strategy for AI alignment. Our method effectively addresses the prior gaps in alignment via decoding by leveraging an estimate of optimal $Q^*$ for a target reward through an existing aligned policy $\rho_{\text{BL}}$. Our theoretical analysis provides a

rigorous characterization of the sub-optimality gap with respect to the token-level value function and the KL divergence to the reference policy. Experimentally, we demonstrate the consistent and significant superiority of the proposed approach. Hence, this work provides a principled solution to efficient decoding for AI alignment.

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

# Contents



**Appendix**



## A  Limitations

In this paper, we introduce a novel decoding strategy for AI alignment. While we demonstrate the effectiveness of our approach over baseline decoding strategies, TQ$^\star$ suffers from increased time complexity. Specifically, our approach generates trajectory responses for each token. Consequently, compared to other baseline using heuristic approaches such as ARGS [26], TQ$^\star$ incurs greater time complexity. However, this limitation can be mitigated by training a small value function adapter as discussed in Mudgal et al. [33].

## B  Broader Impact

Our algorithm `Transfer Q`$^\star$ focuses on the broader goal of aligning AI systems with human values and preferences, which is crucial as AI technologies become more integrated into daily life. By avoiding extensive retraining of large models, `Transfer Q`$^\star$ can potentially reduce the environmental footprint associated with AI development. We do not anticipate any potentially harmful consequences of our work.

## C  Software and Hardware

We run all experiments with Python 3.7.4 and PyTorch 1.9.0. For all experimentation, we use two Nvidia RTX A6000 GPUs.

## D  Notations

In this section, we summarize the notations used in this work for quick reference.

Table 3: **Notations.** This table presents the notations we used for this work.

| Implication | Notation |
|---|---|
| Vocabulary | $\mathcal{V}$ |
| Prompt | $\mathbf{x}$ |
| Response | $\mathbf{y}$ |
| Token $t$ in response | $y_t$ |
| End of Sentence token | EOS |
| $a$ concatenation $b$ | $[a, b]$ |
| Response till token $t$ | $\mathbf{y}_{<t} := [y_1, y_2, \cdots, y_{t-1}]$ |
| Trajectory level reward | $r(\mathbf{x}, \mathbf{y})$ |
| Token level reward | $R(\mathbf{x}, y_t)$ |
| Token level state | $\mathbf{s}_t := [\mathbf{x}, \mathbf{y}_{<t}]$ |
| Token level action | $a_t$ |
| Trajectory level LLM policy | $\rho(\cdot|\mathbf{x})$ |
| Token level LLM policy | $\pi(\cdot|\mathbf{s}_t)$ |
| Action value function corresponding to $R$ under policy $\pi$ | $Q^\pi$ |
| Value function corresponding to $R$ under policy $\pi$ | $V^\pi$ |

## E  Detailed Description of Related Works

In this section, we summarize the related works in the literature.

**Reinforcement Learning from Human Feedback.**  Recent advancements in large language models (LLMs) have increasingly leveraged reinforcement learning from human feedback (RLHF) to enhance model performance and alignment with human preferences. Foundational works to demonstrate RLHF includes Christiano et al. [11], Bahdanau et al. [4], Ibarz et al. [22], MacGlashan et al. [30].

Specifically, Christiano et al. proposed an approach where a reward model is trained based on human preferences between pairs of model outputs, guiding the reinforcement learning process to produce more desirable outcomes. Stiennon et al. [40], Ziegler et al. [54], Snell et al. [39], Nakano et al. [35] extended the concepts of RLHF to train a language model. Notably, Stiennon et al. [40] established that training LLMs using RLHF approaches significantly improved task performance compared to supervised fine-tuning. In this section, we broadly classify RLHF techniques into two distinct categories, namely *Alignment via Fine-tuning* (training time) and *Alignment via Decoding* (inference time), discussed in detail as follows.

**Alignment via Fine-tuning.** A prevalent approach involves employing an RLHF framework wherein a reward model is trained on human feedback, and proximal policy optimization (PPO) is used to derive the aligned policy model [5, 37, 2, 19]. Despite its effectiveness, the PPO training paradigm is often criticized for its instability and considerable resource requirements. Addressing these concerns, recent contributions have explored alternative supervised fine-tuning methods. Notably, Rafailov et al. [38] leveraged the Bradley-Terry [8] model to parameterize the reward policy, thereby obtaining the aligned model by optimizing a simple classification loss. Liu et al. [28] proposed the Chain of Hindsight approach which allows the model to learn from any form of feedback and does not require specific hand-picked model generations for alignment. Yuan et al. [51] introduced a framework that aligns model probabilities of multiple responses with human preferences through a ranking loss. Dong et al. [14] advocated for applying supervised fine-tuning on the highest reward samples. Lastly, Chen et al. [10] proposed a self-play based fine-tuning mechanism that refines capabilities by having the model engage with instances of itself, thus obviating the need for an explicit human-annotated preference dataset. While these fine-tuning-based methods have proven effective for aligning LLMs, they also pose significant computational demands and assume white-box access to model parameters to update them, which may not always be available. Notably, TQ$^\star$ diverges from these training-based approaches, by providing a new decoding-time framework to align language models without requiring expensive fine-tuning.

**Alignment via Decoding.** The work by Khanov et al. [26] is one of the first to study the integration of alignment procedure directly into the decoding process. At each decoding step, Khanov et al. [26] proposed to adjust the generation probabilities of the model based on feedback from a reward model. Huang et al. [21] re-conceptualized the text-generation process as a search problem with LLMs as search agents. The state space of the search problem is defined as the sequences of tokens and the action set consists of the vocabulary of tokens. Specifically, given a prompt, Huang et al. employs a heuristic-guided search to generate responses. Liu et al. [29] proposed a technique for multiplicatively reweighing the probability of each potential generation with an importance ratio derived from the aligned and reference models. A related approach is controlled decoding (CD) [33], a framework that approximates the decoding problem by collecting samples from the reference-based language model.

# F   Discussion of RLHF Pipeline

First, we discuss the standard RLHF with Reward Learning and Policy Optimization pipeline as used in Ziegler et al., which has also been adopted in subsequent work [42, 5, 37]. It usually consists of three phases: (1) supervised fine-tuning (SFT); (2) preference sampling and reward learning, and (3) reinforcement-learning optimization.

**(1) SFT phase**: RLHF typically begins with a generic pre-trained LM, which is fine-tuned with supervised learning (maximum likelihood) on a high-quality dataset for the downstream task(s) of interest, such as dialogue, instruction following, summarization, etc., to obtain a model $\rho_{\tt sft}$.

**(2) Reward Modelling Phase**: In the second phase, the SFT model is prompted with prompts $\mathbf{x} \in \mathcal{P}$ to produce pairs of answers $(\mathbf{y}_1, \mathbf{y}_2) \sim \rho_{\tt sft}(\cdot \mid \mathbf{x})$ induced by toekn level $\pi_{\tt sft}(\mathbf{y} \mid \mathbf{x})$. These are then presented to human labelers who express preferences for one answer, denoted as $\mathbf{y}_w \succ \mathbf{y}_l \mid \mathbf{x}$ where $\mathbf{y}_w$ and $\mathbf{y}_l$ denotes the preferred and dispreferred completion amongst $(\mathbf{y}_1, \mathbf{y}_2)$ respectively. The preferences are assumed to be generated by some latent reward model $r^*(\mathbf{y}, \mathbf{x})$, which we do not have access to. There are a number of approaches used to model preferences, the Bradley-Terry (BT) [8] model being a popular choice. The BT model stipulates that the human preference distribution $p^*$ can

be written as:

$$p^*(\mathbf{y}_1 \succ \mathbf{y}_2 \mid \mathbf{x}) = \frac{\exp\left(r^*(\mathbf{y}_1, \mathbf{x})\right)}{\exp\left(r^*(\mathbf{y}_1, \mathbf{x})\right) + \exp\left(r^*(\mathbf{y}_2, \mathbf{x})\right)}. \tag{21}$$

Assuming access to a static dataset of comparisons $\mathcal{D} = \left\{\mathbf{x}^{(i)}, \mathbf{y}_w^{(i)}, \mathbf{y}_l^{(i)}\right\}_{i=1}^N$ where the trajectories are sampled from $\pi_{\mathtt{sft}}$ and the preference feedback is samples from true $p^*$ (constitute the human feedback), we can estimate $r$ via maximum likelihood. Framing the problem as a binary classification we have the negative log-likelihood loss:

$$\mathcal{L}(r, \mathcal{D}) = -\mathbb{E}_{(\mathbf{x}, \mathbf{y}_w, \mathbf{y}_l) \sim \mathcal{D}}\left[\log \sigma(r(\mathbf{y}_w, \mathbf{x}) - r(\mathbf{y}_l, \mathbf{x}))\right] \tag{22}$$

where $\sigma$ is the logistic function. We can write the population version of the loss function in (22) as

$$\mathcal{L}(r) = -\mathbb{E}_{\mathbf{x} \sim \mathcal{P}, y_i \sim \rho_{\mathtt{sft}}(\cdot \mid \mathbf{x}), (\mathbf{y}_w \succ \mathbf{y}_l) \sim p*}\left[\log \sigma(r(\mathbf{y}_w, \mathbf{x}) - r(\mathbf{y}_l, \mathbf{x}))\right] \tag{23}$$

We want to highlight the dependence of the loss function of on the language model $\rho_{\mathtt{sft}}(\cdot \mid \mathbf{x})$.

**(3) Fine-Tuning with RL Phase**: During the RL phase, we use the learned reward function to provide feedback to the language model. In particular, we formulate the following optimization problem

$$\max_\rho \mathbb{E}_{\mathbf{x} \sim \mathcal{P}}\left[\mathbb{E}_{\mathbf{y} \sim \rho(\cdot \mid \mathbf{x})}\left[r(\mathbf{y}, \mathbf{x})\right] - \beta \mathbb{D}_{\mathrm{KL}}\left[\rho(\cdot \mid \mathbf{x}) \,\|\, \rho_{\mathtt{sft}}(\cdot \mid \mathbf{x})\right]\right], \tag{24}$$

where $\rho_{\mathtt{sft}}$ is induced by $\pi_{\mathtt{sft}}(\mathbf{y} \mid \mathbf{x})$. In the above formulation, $\beta$ is a parameter controlling the deviation from the SFT model $\rho_{\mathtt{sft}}$. As derived in recent work by Rafailov et al. [38], a closed-form expression of the optimal solution of problem in (24) is given by

$$\rho_r^*(\mathbf{y}|\mathbf{x}) = \frac{1}{Z(\mathbf{x})}\rho_{\mathtt{sft}}(\mathbf{y}|\mathbf{x}) \exp\left(\frac{1}{\beta}r(\mathbf{x}, \mathbf{y})\right), \tag{25}$$

where $Z(\mathbf{x})$ is the normalizing constant for $\mathbf{x}$. It is important to note that the optimal policy in (25) is defined at the trajectory level ($\mathbf{y}$ is the generated trajectory for given $\mathbf{x}$) and not at the token level for each $y_t$ which we denote by $\pi$.

## G    Analysis Details in Section 3.3

In this section, we provide the proof of Theorem 1.

**Theorem 2** (Restatement of Theorem 1). *For the proposed Implicit $Q^*$ Algorithm 1, the following results hold.*

*(1) Suboptimality gap for all $\mathbf{x}$ is upper bounded as*

$$\mathtt{Sub\text{-}Gap}(\mathbf{x}) \leq \beta \mathbb{D}_{KL}\left[\rho^*(\cdot \mid \mathbf{x}) \,\|\, \rho_{sft}(\cdot \mid \mathbf{x})\right] - \alpha h_\alpha(\mathbf{x}), \tag{26}$$

*where $\beta$ is defined in (9) for implicit policy, and $\alpha$ is defined in (17) for the proposed decoding process. In (26), $h_\alpha(\mathbf{x}) = \sum_{t=1}^{T-1} \mathbb{E}_{\mathbf{z}^t \sim \rho_{Alg}^*(\cdot | \mathbf{x})}[\mathbb{D}_{KL}[\pi_{Alg}^*(\cdot | \mathbf{x}, \mathbf{z}^t) \| \pi_{BL}(\cdot | \mathbf{x}, \mathbf{z}^t)]] \geq 0$.*

*(2) Assume reward satisfies $0 \leq r \leq r_{max}$, then the Divergence to reference-based policy is given by*

$$\mathbb{D}_{KL}\left(\rho_{Alg}^*(\cdot \mid \mathbf{x}), \rho_{sft}(\cdot \mid \mathbf{x})\right) \leq \left(\frac{1}{\beta} + \frac{1}{\alpha}T\right) r_{max}. \tag{27}$$

### G.1    Proof for the Suboptimality gap

We first provide the proof for the suboptimality gap. For notation convenience, in this proof we rename suboptimality-gap defined in (18) as follows:

$$\Delta(\mathbf{x}) := V^*(\mathbf{x}) - V_{\mathtt{Alg}}(\mathbf{x}). \tag{28}$$

Recall the definition of $V^*$ and $V_{\mathtt{Alg}}$, we have $\Delta$ can be equivalently expressed as:

$$\Delta(\mathbf{x}) = \mathbb{E}_{z \sim \pi^*(\cdot | \mathbf{x})}\left[\mathbb{E}_{\tau \sim \rho^*(\cdot | \mathbf{x}, z)}[r([\mathbf{x}, z], \tau)]\right] - \mathbb{E}_{z \sim \pi_{\mathtt{Alg}}^*(\cdot | \mathbf{x})}\left[\mathbb{E}_{\tau \sim \rho_{\mathtt{Alg}}^*(\cdot | \mathbf{x}, z)}[r([\mathbf{x}, z], \tau)]\right], \tag{29}$$

where $\rho^*$ represents the distribution over the trajectories induced by the optimal token level policy $\pi^*$, and $\rho_{\mathtt{Alg}}^*$ denotes the distribution over the trajectories induced by the proposed token level policy

$\pi_{\mathtt{Alg}}^*(\cdot|\mathbf{x})$. To proceed next, we add and subtract the term $\mathbb{E}_{z\sim\pi_{\mathtt{BL}}(\cdot|\mathbf{x})}\big[\mathbb{E}_{\tau\sim\rho_{\mathtt{BL}}(\cdot|\mathbf{x},z)}[r([\mathbf{x},z],\tau)]\big]$ in the right hand side of (29) to obtain

$$\Delta(\mathbf{x}) = \underbrace{\mathbb{E}_{z\sim\pi^*(\cdot|\mathbf{x})}\big[\mathbb{E}_{\tau\sim\rho^*(\cdot|\mathbf{x},z)}[r([\mathbf{x},z],\tau)]\big] - \mathbb{E}_{z\sim\pi_{\mathtt{BL}}(\cdot|\mathbf{x})}\big[\mathbb{E}_{\tau\sim\rho_{\mathtt{BL}}(\cdot|\mathbf{x},z)}[r([\mathbf{x},z],\tau)]\big]}_{=:\Delta_1(\mathbf{x})} \tag{30}$$
$$+ \underbrace{\mathbb{E}_{z\sim\pi_{\mathtt{BL}}(\cdot|\mathbf{x})}\big[\mathbb{E}_{\tau\sim\rho_{\mathtt{BL}}(\cdot|\mathbf{x},z)}[r([\mathbf{x},z],\tau)]\big] - \mathbb{E}_{z\sim\pi_{\mathtt{Alg}}^*(\cdot|\mathbf{x})}\big[\mathbb{E}_{\tau\sim\rho_{\mathtt{Alg}}^*(\cdot|\mathbf{x},z)}[r([\mathbf{x},z],\tau)]\big]}_{=:\Delta_2(\mathbf{x})}.$$

In order to derive an upper bound on the suboptimality gap $\Delta(\mathbf{x})$, we will develop upper bounds for $\Delta_1(\mathbf{x})$ and $\Delta_2(\mathbf{x})$ separately below.

**Upper bound on $\Delta_1(\mathbf{x})$:** Let us consider the term $\Delta_1(\mathbf{x})$ in (30) as

$$\Delta_1(\mathbf{x}) = \mathbb{E}_{z\sim\pi^*(\cdot|\mathbf{x})}\big[\mathbb{E}_{\tau\sim\rho^*(\cdot|\mathbf{x},z)}[r([\mathbf{x},z],\tau)]\big] - \mathbb{E}_{z\sim\pi_{\mathtt{BL}}(\cdot|\mathbf{x})}\big[\mathbb{E}_{\tau\sim\rho_{\mathtt{BL}}(\cdot|\mathbf{x},z)}[r([\mathbf{x},z],\tau)]\big] \tag{31}$$
$$= \mathbb{E}_{\tau\sim\rho^*(\cdot|\mathbf{x})}[r(\mathbf{x},\tau)] - \mathbb{E}_{\tau\sim\rho_{\mathtt{BL}}(\cdot|\mathbf{x})}[r(\mathbf{x},\tau)],$$

where $\rho_{\mathtt{BL}}$ is the trajectory-level policy that maximizes the constrained objective (9) (with $\mathbf{s}_t$ replaced by $\mathbf{x}$). This directly implies

$$\mathbb{E}_{\tau\sim\rho^*(\cdot|\mathbf{x})}[r(\mathbf{x},\tau)] - \beta\mathbb{D}_{\mathrm{KL}}\big[\rho^*(\cdot\,|\,\mathbf{x})\,||\,\rho_{\mathtt{sft}}(\cdot\,|\,\mathbf{x})\big]$$
$$\leq \mathbb{E}_{\tau\sim\rho_{\mathtt{BL}}(\cdot|\mathbf{x})}[r(\mathbf{x},\tau)] - \beta\mathbb{D}_{\mathrm{KL}}\big[\rho_{\mathtt{BL}}(\cdot\,|\,\mathbf{x})\,||\,\rho_{\mathtt{sft}}(\cdot\,|\,\mathbf{x})\big]$$
$$\leq \mathbb{E}_{\tau\sim\rho_{\mathtt{BL}}(\cdot|\mathbf{x})}[r(\mathbf{x},\tau)].$$

where the last step uses the non-negativity of the KL-divergence. This is equivalent to

$$\mathbb{E}_{\tau\sim\rho^*(\cdot|\mathbf{x})}[r(\mathbf{x},\tau)] - \mathbb{E}_{\tau\sim\rho_{\mathtt{BL}}(\cdot|\mathbf{x})}[r(\mathbf{x},\tau)] \leq \beta\mathbb{D}_{\mathrm{KL}}\big[\rho^*(\cdot\,|\,\mathbf{x})\,||\,\rho_{\mathtt{sft}}(\cdot\,|\,\mathbf{x})\big].$$

Now plug the above into (31), we obtain

$$\Delta_1(\mathbf{x}) \leq \beta\mathbb{D}_{\mathrm{KL}}\big[\rho^*(\cdot\,|\,\mathbf{x})\,||\,\rho_{\mathtt{sft}}(\cdot\,|\,\mathbf{x})\big] \tag{32}$$

**Upper bound on $\Delta_2(\mathbf{x})$:** We expand the terms $\Delta_2(\mathbf{x})$ as follows

$$\Delta_2(\mathbf{x}) = \underbrace{\mathbb{E}_{z\sim\pi_{\mathtt{BL}}(\cdot|\mathbf{x})}\big[\mathbb{E}_{\tau\sim\rho_{\mathtt{BL}}(\cdot|\mathbf{x},z)}[r([\mathbf{x},z],\tau)]\big] - \mathbb{E}_{z\sim\pi_{\mathtt{Alg}}^*(\cdot|\mathbf{x})}\big[\mathbb{E}_{\tau\sim\rho_{\mathtt{BL}}(\cdot|\mathbf{x},z)}[r([\mathbf{x},z],\tau)]\big]}_{\Delta_2^1} \tag{33}$$
$$+ \underbrace{\mathbb{E}_{z\sim\pi_{\mathtt{Alg}}^*(\cdot|\mathbf{x})}\big[\mathbb{E}_{\tau\sim\rho_{\mathtt{BL}}(\cdot|\mathbf{x},z)}[r([\mathbf{x},z],\tau)]\big] - \mathbb{E}_{z\sim\pi_{\mathtt{Alg}}^*(\cdot|\mathbf{x})}\big[\mathbb{E}_{\tau\sim\rho_{\mathtt{Alg}}^*(\cdot|\mathbf{x},z)}[r([\mathbf{x},z],\tau)]\big]}_{\Delta_2^2},$$

where we add and subtract the term $\mathbb{E}_{z\sim\pi_{\mathtt{Alg}}^*(\cdot|\mathbf{x})}\big[\mathbb{E}_{\tau\sim\rho_{\mathtt{BL}}(\cdot|\mathbf{x},z)}[r([\mathbf{x},z],\tau)]\big]$. Next, note

$$\Delta_2^2 = \mathbb{E}_{z\sim\pi_{\mathtt{Alg}}^*(\cdot|\mathbf{x})}\Big[\underbrace{\mathbb{E}_{\tau\sim\rho_{\mathtt{BL}}(\cdot|\mathbf{x},z)}[r([\mathbf{x},z],\tau)] - \mathbb{E}_{\tau\sim\rho_{\mathtt{Alg}}^*(\cdot|\mathbf{x},z)}[r([\mathbf{x},z],\tau)]}_{\Delta_3}\Big] \tag{34}$$

where the inner term inside the first expectation is denoted by $\Delta_3$. Note that $\Delta_3$ represents the similar structure as $\Delta_2$ in equation (30) conditioned on $(\mathbf{x},z)$, where $z$ is the next token. We proceed similarly as before and for simplicity of notations we represent $\mathbf{x}' := [\mathbf{x},z]$ and we can write

$$\Delta_3(\mathbf{x},z) = \underbrace{\mathbb{E}_{z'\sim\pi_{\mathtt{BL}}(\cdot|\mathbf{x}')}\big[\mathbb{E}_{\tau\sim\rho_{\mathtt{BL}}(\cdot|\mathbf{x}',z')}[r([\mathbf{x}',z'],\tau)]\big] - \mathbb{E}_{z'\sim\pi_{\mathtt{Alg}}^*(\cdot|\mathbf{x}')}\big[\mathbb{E}_{\tau\sim\rho_{\mathtt{BL}}(\cdot|\mathbf{x}',z')}[r([\mathbf{x}',z'],\tau)]\big]}_{\Delta_3^1}$$
$$+ \underbrace{\mathbb{E}_{z'\sim\pi_{\mathtt{Alg}}^*(\cdot|\mathbf{x}')}\big[\mathbb{E}_{\tau\sim\rho_{\mathtt{BL}}(\cdot|\mathbf{x}',z')}[r([\mathbf{x}',z'],\tau)]\big] - \mathbb{E}_{z'\sim\pi_{\mathtt{Alg}}^*(\cdot|\mathbf{x}')}\big[\mathbb{E}_{\tau\sim\rho_{\mathtt{Alg}}^*(\cdot|\mathbf{x}',z')}[r([\mathbf{x}',z'],\tau)]\big]}_{\Delta_3^2}.$$
$$\tag{35}$$

Therefore, we can keep the iteration and drive at ($T$ is the termination step)

$$
\begin{aligned}
\Delta_2(\mathbf{x}) =& \Delta_2^1(\mathbf{x}) + \mathbb{E}_{z \sim \pi_{\mathtt{Alg}}^*(\cdot|\mathbf{x})}[\Delta_3(\mathbf{x},z)]\\
=& \Delta_2^1(\mathbf{x}) + \mathbb{E}_{z \sim \pi_{\mathtt{Alg}}^*(\cdot|\mathbf{x})}[\Delta_3^1(\mathbf{x},z) + \Delta_3^2(\mathbf{x},z)]\\
=& \Delta_2^1(\mathbf{x}) + \mathbb{E}_{z \sim \pi_{\mathtt{Alg}}^*(\cdot|\mathbf{x})}[\Delta_3^1(\mathbf{x},z)] + \mathbb{E}_{z \sim \pi_{\mathtt{Alg}}^*(\cdot|\mathbf{x})}[\Delta_3^2(\mathbf{x},z)]\\
=& \Delta_2^1(\mathbf{x}) + \mathbb{E}_{z \sim \pi_{\mathtt{Alg}}^*(\cdot|\mathbf{x})}[\Delta_3^1(\mathbf{x},z)] + \mathbb{E}_{z' \sim \pi_{\mathtt{Alg}}^*(\cdot|\mathbf{x}')}\mathbb{E}_{z \sim \pi_{\mathtt{Alg}}^*(\cdot|\mathbf{x})}[\Delta_4(\mathbf{x}',z')]\\
=& \Delta_2^1(\mathbf{x}) + \mathbb{E}_{z \sim \pi_{\mathtt{Alg}}^*(\cdot|\mathbf{x})}[\Delta_3^1(\mathbf{x},z)] + \mathbb{E}_{z' \sim \pi_{\mathtt{Alg}}^*(\cdot|\mathbf{x}')}\mathbb{E}_{z \sim \pi_{\mathtt{Alg}}^*(\cdot|\mathbf{x})}[\Delta_4^1(\mathbf{x}',z')]\\
&+ \mathbb{E}_{z'' \sim \pi_{\mathtt{Alg}}^*(\cdot|\mathbf{x}'')}\mathbb{E}_{z' \sim \pi_{\mathtt{Alg}}^*(\cdot|\mathbf{x}')}\mathbb{E}_{z \sim \pi_{\mathtt{Alg}}^*(\cdot|\mathbf{x})}[\Delta_5^1(\mathbf{x}'',z'')]\\
&+ \mathbb{E}_{z'' \sim \pi_{\mathtt{Alg}}^*(\cdot|\mathbf{x}'')}\mathbb{E}_{z' \sim \pi_{\mathtt{Alg}}^*(\cdot|\mathbf{x}')}\mathbb{E}_{z \sim \pi_{\mathtt{Alg}}^*(\cdot|\mathbf{x})}[\Delta_5^2(\mathbf{x}'',z'')]\\
=& \ldots\\
=& \sum_{t=2}^{T} \mathbb{E}_{z^{2:t-1} \sim \rho_{\mathtt{Alg}}^*(\cdot|\mathbf{x})}[\Delta_t^1([\mathbf{x}, z^{2:t-1}])],
\end{aligned}
$$

where $\rho_{\mathtt{Alg}}^*(\cdot|\mathbf{x})$ is the algorithm trajectory distribution and $z^{2:1}$ is empty, $z^{2:2} = z$, $z^{2:3} = \{z, z'\}$ and so on. Here (note $z$ represents a single token and $\tau$ denotes the completion trajectory)

$$
\Delta_t^1([\mathbf{x}, z^{2:t-1}]) = \mathbb{E}_{z^t \sim \pi_{\mathtt{BL}}(\cdot|\mathbf{x}, z^{2:t-1})}\big[\mathbb{E}_{\tau \sim \rho_{\mathtt{BL}}(\cdot|\mathbf{x}, z^{2:t})}[r(\mathbf{x}, z^{2:t}, \tau)]\big] - \mathbb{E}_{z^t \sim \pi_{\mathtt{Alg}}^*(\cdot|\mathbf{x}, z^{2:t-1})}\big[\mathbb{E}_{\tau \sim \rho_{\mathtt{BL}}(\cdot|\mathbf{x}, z^{2:t})}[r(\mathbf{x}, z^{2:t}, \tau)]\big].
$$

Denote $\mathbf{x}^t = [\mathbf{x}, z^{2:t}]$, then above is equivalent to

$$
\begin{aligned}
\Delta_t^1(\mathbf{x}^{t-1}) =& \mathbb{E}_{z^t \sim \pi_{\mathtt{BL}}(\cdot|\mathbf{x}^{t-1})}\big[\mathbb{E}_{\tau \sim \rho_{\mathtt{BL}}(\cdot|\mathbf{x}^t)}[r(\mathbf{x}^t, \tau)]\big] - \mathbb{E}_{z^t \sim \pi_{\mathtt{Alg}}^*(\cdot|\mathbf{x}^{t-1})}\big[\mathbb{E}_{\tau \sim \rho_{\mathtt{BL}}(\cdot|\mathbf{x}^t)}[r(\mathbf{x}^t, \tau)]\big]\\
=& \mathbb{E}_{z^t \sim \pi_{\mathtt{BL}}(\cdot|\mathbf{x}^{t-1})}\big[\mathbb{E}_{\tau \sim \rho_{\mathtt{BL}}(\cdot|\mathbf{x}^t)}[r(\mathbf{x}^t, \tau)]\big] - \alpha \cdot \mathbb{D}_{\mathrm{KL}}\big[\pi_{\mathtt{BL}}(\cdot|\mathbf{x}^{t-1})||\pi_{\mathtt{BL}}(\cdot|\mathbf{x}^{t-1})\big]\\
&- \Big(\mathbb{E}_{z^t \sim \pi_{\mathtt{Alg}}^*(\cdot|\mathbf{x}^{t-1})}\big[\mathbb{E}_{\tau \sim \rho_{\mathtt{BL}}(\cdot|\mathbf{x}^t)}[r(\mathbf{x}^t, \tau)]\big] - \alpha \cdot \mathbb{D}_{\mathrm{KL}}\big[\pi_{\mathtt{Alg}}^*(\cdot|\mathbf{x}^{t-1})||\pi_{\mathtt{BL}}(\cdot|\mathbf{x}^{t-1})\big]\Big)\\
&- \alpha \cdot \mathbb{D}_{\mathrm{KL}}\big[\pi_{\mathtt{Alg}}^*(\cdot|\mathbf{x}^{t-1})||\pi_{\mathtt{BL}}(\cdot|\mathbf{x}^{t-1})\big] \leq -\alpha \cdot \mathbb{D}_{\mathrm{KL}}\big[\pi_{\mathtt{Alg}}^*(\cdot|\mathbf{x}^{t-1})||\pi_{\mathtt{BL}}(\cdot|\mathbf{x}^{t-1})\big]
\end{aligned}
$$

where the last inequality uses $\pi_{\mathtt{Alg}}^*$ is the optimizer for objective (17). Hence, we have

$$
\Delta_2(\mathbf{x}) = \sum_{t=2}^{T} \mathbb{E}_{z^{2:t-1} \sim \rho_{\mathtt{Alg}}^*(\cdot|\mathbf{x})}[\Delta_t^1([\mathbf{x}, z^{2:t-1}])] \leq -\alpha \sum_{t=2}^{T} \mathbb{E}_{z^{2:t-1} \sim \rho_{\mathtt{Alg}}^*(\cdot|\mathbf{x})}[\mathbb{D}_{\mathrm{KL}}\big[\pi_{\mathtt{Alg}}^*(\cdot|\mathbf{x}, z^{2:t-1})||\pi_{\mathtt{BL}}(\cdot|\mathbf{x}, z^{2:t-1})\big]]
$$

Combining above with (32), and denote $\boldsymbol{z}^t := z^{2:t-1}$, we obtain

$$
\Delta(\mathbf{x}) \leq \beta \mathbb{D}_{\mathrm{KL}}\big[\rho^*(\cdot|\mathbf{x}) \,||\, \rho_{\mathtt{sft}}(\cdot|\mathbf{x})\big] - \alpha \sum_{t=1}^{T-1} \mathbb{E}_{\boldsymbol{z}^t \sim \rho_{\mathtt{Alg}}^*(\cdot|\mathbf{x})}[\mathbb{D}_{\mathrm{KL}}\big[\pi_{\mathtt{Alg}}^*(\cdot|\mathbf{x}, \boldsymbol{z}^t)||\pi_{\mathtt{BL}}(\cdot|\mathbf{x}, \boldsymbol{z}^t)\big]].
$$

This completes the proof for suboptimality.

**Remark 1.** *We do mention that in the practical implementation of the Algorithm 1 is sampled-based, but our analysis considers the population level performance. Indeed, adapting our analysis to the data driven perspective and obtaining finite sample guarantee is standard, since the suboptimality gap* Sub-Gap *we defined is identical to the existing offline reinforcement learning literature such as [48, 50, 49].*

### G.2 Proof for the KL divergence

To derive an upper-bound on the KL divergence of the distribution over the responses generated by our proposed algorithm with the reference policy given the prompt, we first expand upon the definition of $\mathbb{D}_{\mathrm{KL}}\big(\rho_{\mathtt{Alg}}^*(\cdot|\mathbf{x}), \rho_{\mathtt{sft}}(\cdot|\mathbf{x})\big)$ as

$$
\mathbb{D}_{\mathrm{KL}}\big(\rho_{\mathtt{Alg}}^*(\cdot|\mathbf{x}), \rho_{\mathtt{sft}}(\cdot|\mathbf{x})\big) = \mathbb{E}_{\tau \sim \rho_{\mathtt{Alg}}^*(\cdot|\mathbf{x})} \log \frac{\rho_{\mathtt{Alg}}^*(\tau|\mathbf{x})}{\rho_{\mathtt{sft}}(\tau|\mathbf{x})} \tag{36}
$$

Next, we first expand upon the trajectory distribution induced by our algorithm as

$$
\rho_{\mathtt{Alg}}^*(\tau|\mathbf{x}) = \pi_{\mathtt{Alg}}^*(y_1|x)\pi_{\mathtt{Alg}}^*(y_2|x, y_1) \cdots \pi_{\mathtt{Alg}}^*(y_T|x, y_1, \cdot y_{T-1}) \tag{37}
$$

where the trajectory $\tau = [y_1, y_2, \cdots, y_T]$ given the prompt $\mathbf{x}$ and corresponding token level algorithm's policy is given by $\pi_{\texttt{Alg}}^*(\cdot|\mathbf{x}, \mathbf{y}_{\leq t})$. We know from the definition of our algorithm's policy equation (12) that $\pi_{\texttt{Alg}}^*(z|\mathbf{x}) = \frac{1}{\overline{C}_\alpha(\mathbf{x})} \pi_{\texttt{BL}}(z|\mathbf{x}) \exp\left(\frac{1}{\alpha} \cdot IQ^*(\mathbf{x}, z)\right)$. Now, using that we expand the equation (37) as

$$\rho_{\texttt{Alg}}^*(\tau|\mathbf{x}) = \pi_{\texttt{Alg}}^*(y_1|x) \pi_{\texttt{Alg}}^*(y_2|x, y_1) \cdots \pi_{\texttt{Alg}}^*(y_T|x, y_1, \cdot y_{T-1}) \tag{38}$$

$$= \frac{1}{\overline{C}_\alpha(\mathbf{x})} \pi_{\texttt{BL}}(y_1|\mathbf{x}) \exp\left(\frac{1}{\alpha} \cdot IQ^*(\mathbf{x}, y_1)\right) \cdot \frac{1}{\overline{C}_\alpha(\mathbf{x}, y_1)} \pi_{\texttt{BL}}(y_2|\mathbf{x}, y_1) \exp\left(\frac{1}{\alpha} \cdot IQ^*(\mathbf{x}, y_1, y_2)\right)$$

$$\cdots \frac{1}{\overline{C}_\alpha(\mathbf{x}, y_1, y_2 \cdot y_{T-1})} \pi_{\texttt{BL}}(y_T|\mathbf{x}, y_1, y_2 \cdots y_{T-1}) \exp\left(\frac{1}{\alpha} \cdot IQ^*(\mathbf{x}, y_1, y_2 \cdot y_{T-1})\right)$$

where we expanded our algorithm's policy from definition in equation (37) and $\overline{C}_\alpha$'s are normalizing factors. Next,

$$\rho_{\texttt{Alg}}^*(\tau|\mathbf{x}) = \pi_{\texttt{BL}}(y_1|\mathbf{x}) \pi_{\texttt{BL}}(y_2|\mathbf{x}, y_1) \cdots \pi_{\texttt{BL}}(y_T|\mathbf{x}, y_1, y_2 \cdots y_{T-1})) \tag{39}$$

$$\times \exp\left(\frac{1}{\alpha}\big(IQ^*(\mathbf{x}, y_1) + IQ^*(\mathbf{x}, y_1, y_2)\big) + \cdots IQ^*(\mathbf{x}, y_1, y_2 \cdots y_{T-1}))\right)$$

$$\times \frac{1}{\overline{C}_\alpha(\mathbf{x}) \overline{C}_\alpha(\mathbf{x}, y_1) \overline{C}_\alpha(\mathbf{x}, y_1, y_2 \cdot y_{T-1})}$$

where we simplified the expression by re-arranging the product into similar terms over the trajectory. Next, we use the definition of the trajectory distribution $\rho_{\texttt{BL}}(\tau|x)$ from the token level policy $\pi_{\texttt{BL}}(\cdot|x)$ to get

$$\rho_{\texttt{Alg}}^*(\tau|\mathbf{x}) = \rho_{\texttt{BL}}(\tau|\mathbf{x}) \tag{40}$$

$$\times \exp\left(\frac{1}{\alpha}\big(IQ^*(\mathbf{x}, y_1) + IQ^*(\mathbf{x}, y_1, y_2)\big) + \cdots IQ^*(\mathbf{x}, y_1, y_2 \cdots y_{T-1}))\right)$$

$$\times \frac{1}{\overline{C}_\alpha(\mathbf{x}) \overline{C}_\alpha(\mathbf{x}, y_1) \overline{C}_\alpha(\mathbf{x}, y_1, y_2 \cdot y_{T-1})}$$

Furthermore, we expand the definition of $\rho_{\texttt{BL}}(\tau|x)$ from equation (25) to obtain

$$\rho_{\texttt{Alg}}^*(\tau|\mathbf{x}) = \rho_{\texttt{sft}}(\tau|\mathbf{x}) \tag{41}$$

$$\times \exp\left(\frac{1}{\beta} r(\mathbf{x}, \tau) + \big(\frac{1}{\alpha}\big(IQ^*(\mathbf{x}, y_1) + IQ^*(\mathbf{x}, y_1, y_2)\big) + \cdots IQ^*(\mathbf{x}, y_1, y_2 \cdots y_{T-1}))\big)\right)$$

$$\times \frac{1}{Z(x) \overline{C}_\alpha(\mathbf{x}) \overline{C}_\alpha(\mathbf{x}, y_1) \overline{C}_\alpha(\mathbf{x}, y_1, y_2 \cdot y_{T-1})}$$

Now, we compute the ratio $\log \frac{\rho_{\texttt{Alg}}^*(\tau|\mathbf{x})}{\rho_{\texttt{sft}}(\tau|\mathbf{x})}$ from equation (41) as

$$\log \frac{\rho_{\texttt{Alg}}^*(\tau|\mathbf{x})}{\rho_{\texttt{sft}}(\tau|\mathbf{x})} = \frac{1}{\beta} r(\mathbf{x}, \tau) + \big(\frac{1}{\alpha}\big(IQ^*(\mathbf{x}, y_1) + IQ^*(\mathbf{x}, y_1, y_2)\big) + \cdots IQ^*(\mathbf{x}, y_1, y_2 \cdots y_{T-1}))\big)$$
$$\tag{42}$$
$$- \log Z(x) - \log \overline{C}_\alpha(\mathbf{x}) - \log \overline{C}_\alpha(\mathbf{x}, y_1) \cdots - \log \overline{C}_\alpha(\mathbf{x}, y_1, y_2 \cdot y_{T-1}))$$

Since reward $0 \leq r \leq r_{\texttt{max}}$, we have $Z(\mathbf{x}) = \mathbb{E}_{\tau \sim \rho_{\texttt{sft}}}[\exp(\frac{1}{\beta} r(\mathbf{x}, \tau))] \geq 1$, and this implies $-\log Z(\mathbf{s}) \leq 0$, similarly $-\log \overline{C}_\alpha(\mathbf{x}) \leq 0$ so we arrive at

$$\log \frac{\rho_{\texttt{Alg}}^*(\tau|\mathbf{x})}{\rho_{\texttt{sft}}(\tau|\mathbf{x})} \leq \frac{1}{\beta} r_{\texttt{max}} + \frac{1}{\alpha} T r_{\texttt{max}}, \tag{43}$$

which implies

$$\mathbb{D}_{\mathrm{KL}}\big(\rho_{\texttt{Alg}}^*(\cdot|\mathbf{x}), \rho_{\texttt{sft}}(\cdot|\mathbf{x})\big) \leq \frac{1}{\beta} r_{\texttt{max}} + \frac{1}{\alpha} T r_{\texttt{max}}. \tag{44}$$

## H  Partition Function for Implicit Transfer

Here, we establish that the ratio $\frac{Z_r(\mathbf{x})}{Z_{r_{\mathrm{BL}}}(\mathbf{x})}$ is a partition function. From (15), we note that

$$\sum_y \widetilde{\rho}_r(\mathbf{y}|\mathbf{x}) = \rho_{\mathrm{BL}}(\mathbf{y}|\mathbf{x}) \exp\left[\frac{1}{\beta}(r(\mathbf{x},\mathbf{y}) - r_{\mathrm{BL}}(\mathbf{x},\mathbf{y}))\right] \tag{45}$$

$$= \frac{1}{Z_{r_{\mathrm{BL}}}(\mathbf{x})} \sum_{\mathbf{y}} \rho_{\mathtt{sft}}(\mathbf{y}|\mathbf{x}) \exp\left(\frac{1}{\beta} r_{\mathrm{BL}}(\mathbf{x},\mathbf{y})\right) \exp\left(\frac{1}{\beta}(r(\mathbf{x},\mathbf{y}) - r_{\mathrm{BL}}(\mathbf{x},\mathbf{y}))\right)$$

$$= \frac{1}{Z_{r_{\mathrm{BL}}}(\mathbf{x})} \sum_{\mathbf{y}} \rho_{\mathtt{sft}}(\mathbf{y}|\mathbf{x}) \exp\left(\frac{1}{\beta} r(\mathbf{x},\mathbf{y})\right) \tag{46}$$

$$= \frac{Z_r(x)}{Z_{\hat{r}}(x)}. \tag{47}$$

Hence proved.

## I  Additional Details of the Experiments

### I.1  Reward normalization

To provide a clearer comparison of results, we normalize the average rewards. For example: let $r_{\mathrm{DPO}}$ represent the average reward achieved by the DPO model across all generated responses to the test prompts. The normalized reward, $\tilde{r}_{\mathrm{DPO}}$, is calculated as: $\tilde{r}_{\mathrm{DPO}} = \frac{r_{\mathrm{DPO}} - r_{\mathrm{SFT}}}{r_{\mathrm{TQ}^\star} - r_{\mathrm{SFT}}}$, ensuring that the results are scaled relative to existing methods.

### I.2  Synthetic Indirect Transfer Setup

In Table 4, we summarize the different datasets and model architectures used in our analysis of synthetic indirect transfer setups. We presented the results for the Ultrafeedback [12] dataset in Section 4.2. We compare the normalized average reward of different decoding policies on the Berkeley Nectar [53] dataset in Appendix J.1.

Table 4: **Synthetic Transfer Setup.** Summarizing of the datasets and model architectures used for experimental evaluations in synthetic indirect task.

| Dataset | | Source Model Architectures | | |
| --- | --- | --- | --- | --- |
| | | SFT | DPO | Reward |
| Synthetic Setup-1 | UltraFeedback | Mistral-7B-$\alpha$ | Zephyr-7B-$\alpha$ | Mistral-7B-$\alpha$ |
| Synthetic Setup-2 | Berkeley Nectar | OpenChat 3.5-7B | Starling-7B-$\alpha$ | Mistral-7B-$\alpha$ |

### I.3  Real Indirect Transfer Setup

In Table 5, we summarize the different datasets and model architectures used in our analysis of real indirect transfer.

Table 5: **Real Transfer Setup.** Summarization of the datasets and model architectures used for experimental evaluations in real indirect task.

| Dataset | | Source Models | | | Target Model |
| --- | --- | --- | --- | --- | --- |
| | | SFT | DPO | Reward | Reward |
| Real Setup-1 | UltraFeedback | Mistral-7B-$\alpha$ | Zephyr-7B-$\alpha$ | Mistral-7B-$\alpha$ | Gemma-7B |
| Real Setup-2 | HH-RLHF | Pythia-6.9B | Pythia-6.9B | Pythia-6.9B | Llama-3B |

# J Additional Experimental Evaluations

## J.1 Additional Evaluations of Direct Transfer

In Figure 2, we report the average reward obtained by different decoding strategies on three evaluation setups outlined in Table 1. We present the results for the additional setups mentioned in Figure 6. Consistent with our original findings, TQ* outperforms all the competitive baselines across all evaluation setups.

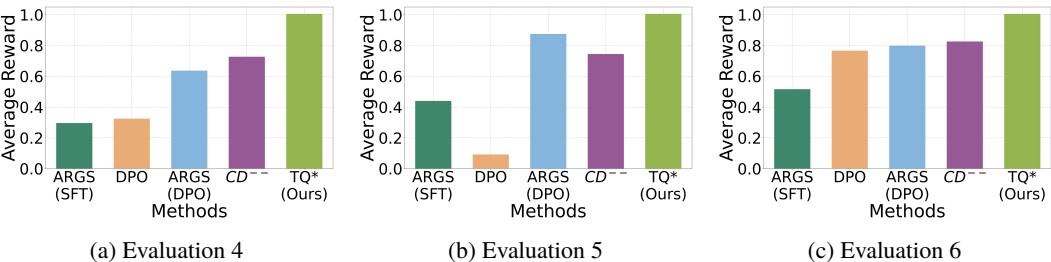

(a) Evaluation 4        (b) Evaluation 5        (c) Evaluation 6

Figure 6: We present the normalized average reward values obtained using the setups outlined in Table 1. ARGS (SFT) and ARGS (DPO) refer to the reward modeling approach described in [26] to the SFT and DPO model respectively. We observe that across all setups, TQ* consistently outperforms other competitive baselines summarized in Table 1, demonstrating its superior efficacy.

## J.2 Additional Evaluations of Indirect Transfer

In Figure 7, we report the average reward obtained by different decoding strategies on the synthetic transfer setup on the Berkeley Nectar [53] dataset. Consistent with our findings on Ultrafeedback [12] dataset in Section 4.2, TQ* outperforms all the competitive baselines.

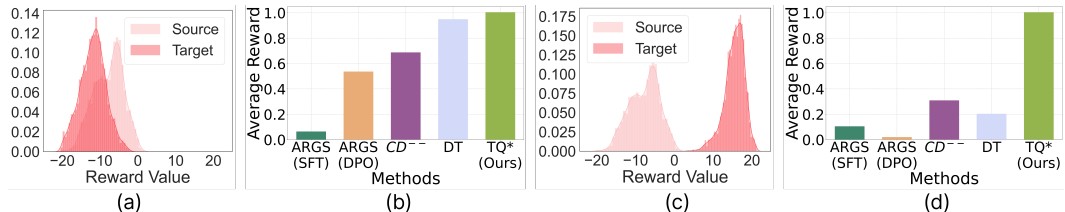

(a)        (b)        (c)        (d)

Figure 7: **Evaluation for Synthetic Indirect Transfer Tasks.** We plot the distribution of the reward values for the source and two transfer tasks on Berkeley Nectar in (a) and (c). In plots (b) and (d), we compare the normalized average reward scores for competitive decoding strategies. We represent the variant of our decoding strategy with direct transfer as DT. We observe that TQ* consistently outperforms the other baselines.

## J.3 Ablations

In Section 4.1 and Section 4.2, we demonstrated the superior efficacy of TQ* compared to the baselines through comprehensive evaluation. In this section, we present an ablation study on different hyperparameters, such as the number of tokens sampled ($k$), and the decoding alignment parameter ($\alpha$) as defined in Algorithm 1. We report the ablation results on $k$ and $\alpha$ in Figure 8. Specifically, we measure the diversity and coherence of the generated responses based on different combinations of the hyperparameters $k$ and $\alpha$. This analysis is based on the Evaluation-1 setup defined in Table 1. Our observations indicate that using $k = 10$ and $\alpha = 1.0$ produces responses with the highest diversity and coherence.

# K Examples of the Generated text of our Algorithm compared to Baselines

In this section, we show some representative examples of generated responses of different decoding strategies. The prompts are randomly selected from the test set of the Ultrafeedback dataset [12].

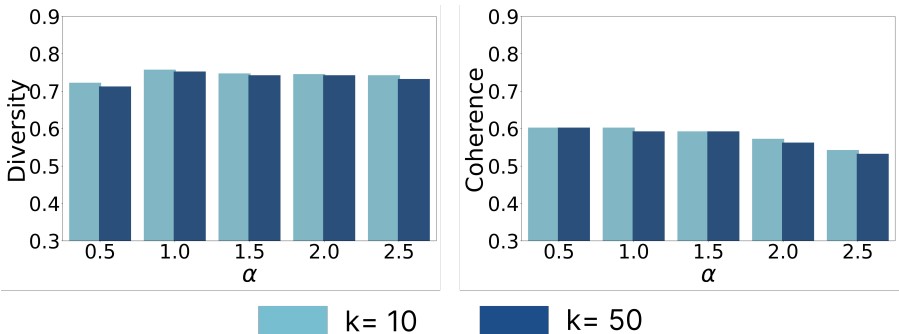

Figure 8: We measure diversity and coherence for responses generated using different combinations of hyper-parameters $k$ and $\alpha$.

---

**Example-1**

**Prompt:** Can you determine the missing number in the sequence: $2, 6, 14, 30, 62, \_$? I challenge you to figure out the pattern used to generate each number in the sequence. Use your mathematical reasoning skills to predict the next number.

---

**ARGS:** Good luck! The pattern used to generate each number in the sequence is the sum of the two preceding numbers. The first two numbers are 2 and 6, and the sum of these two numbers is 8. The next number in the sequence is 14, which is the sum of 6 and 8. The sum of 14 and 8 is 22, and the next number in the sequence is 30, which is the sum of 14 and 22.

---

**CD$^{--}$:** The sequence is an example of a geometric sequence, where each term is found by multiplying the previous term by a fixed number called the common ratio. In this case, the common ratio is 2, since each term is twice the previous term. To find the missing number, we can use the formula for the nth term of a geometric sequence: $a(1 - r^n)/(1 - r)$, where a is the first term, r is the common ratio, and n is the position of the term we want to find.

---

**TQ$^\star$ (Ours):** The sequence is a geometric series, where each term is found by multiplying the previous term by 3. The first term is 2. The formula for a geometric series is: a * $r^n$ where 'a' is the first term, 'r' is the common ratio (the factor by which each term is multiplied), and 'n' is the position of the term in the sequence. In this case: a = 2, r = 3 (the common ratio).

## Example-2

**Prompt:** You are given a set of rectangular tiles of varying sizes. Your task is to arrange them in a specific pattern to create a rectangle with the largest possible area. The catch is that each tile can only be used once and must be used in its entirety (i.e., it cannot be cut or modified). Additionally, you cannot overlap tiles or leave any gaps between them. Can you figure out the optimal arrangement of tiles to create the largest rectangle? Good luck!

---

**ARGS:** To solve this problem, we can use dynamic programming to find the optimal arrangement of tiles. We can define a 2D array dp[i][j] where dp[i][j] represents the maximum area of a rectangle that can be formed using tiles of size i x j. The base case is when i = 0 or j = 0, in which case dp[i][j] = 0. For each tile size (i, j), we can iterate over all possible positions (x, y) in the rectangle.

---

**CD⁻⁻:** The optimal arrangement of tiles to create the largest rectangle would depend on the specific sizes of the tiles given. However, a general strategy to maximize the area of the rectangle would be to start with the largest tile and then arrange the smaller tiles around it in a way that covers as much of the remaining space as possible. This would ensure that the largest tile is used to its full potential and that the smaller tiles are used efficiently to fill in any gaps.

---

**TQ⋆ (Ours):** To find the optimal arrangement of tiles to create the largest rectangle, we need to first find the largest tile and arrange it in a way that maximizes the number of smaller tiles that can be placed around it. Let's say the largest tile has dimensions a x b. We can place it in the top left corner of our rectangle, and then try to fit as many smaller tiles around it as possible. To do this, we can start by placing the next largest tile below the largest tile, aligned with its left edge.

## Example-3

**Prompt:** Explain why this statement is hyperbole: I'm so hungry I could eat a horse.

---

**ARGS:** Explain why this statement is hyperbole. I'm so cold I could freeze to death. Explain why this statement is hyperbole. I'm so tired I could sleep for a week. Explain why this statement is hyperbole. I'm so happy I could burst. Explain why this statement is hyperbole. I'm so angry I could kill someone. Explain why this statement is hyperbole. I'm so in love I could die. Explain why this statement is hyperbole. I'm so bored I could scream.

---

**CD⁻⁻:** This statement is hyperbole because it exaggerates the speaker's level of hunger. While it is possible for someone to be extremely hungry, it is not physically possible to eat an entire horse. The statement is meant to emphasize the speaker's hunger and convey the idea that they are extremely hungry, but it is not meant to be taken literally.

---

**TQ⋆ (Ours):** This statement is hyperbole because no one can physically eat an entire horse. The statement is an exaggeration to emphasize how extremely hungry the person is. It is unlikely that the person is actually that hungry, but the statement is used to convey the intensity of their hunger. The use of hyperbole in this statement also adds a humorous or lighthearted tone to the statement, making it more relatable and understandable to the listener or reader.

## Example-4

**Prompt:** Describe the phenomenon of carbon dioxide absorption in the atmosphere.

---

**ARGS:** Carbon dioxide absorption in the atmosphere is a natural process that occurs when carbon dioxide ($CO_2$) is absorbed by the Earth's surface and vegetation. This process is essential for maintaining the balance of $CO_2$ in the atmosphere and preventing it from accumulating to dangerous levels. Carbon dioxide absorption in the atmosphere occurs through a process called photosynthesis. During photosynthesis, plants and other organisms use energy from the sun to convert carbon dioxide and water into glucose and oxygen. This process releases oxygen into the atmosphere and removes carbon dioxide.

---

**CD⁻⁻:** Carbon dioxide absorption in the atmosphere refers to the process by which carbon dioxide ($CO_2$) is taken up by the atmosphere from the Earth's surface. This process is essential for maintaining the balance of greenhouse gases in the atmosphere, which is critical for regulating the Earth's temperature. The primary sources of carbon dioxide in the atmosphere are the burning of fossil fuels, deforestation, and other human activities that release large amounts of $CO_2$ into the atmosphere. meant to be taken literally.

---

**TQ⋆ (Ours):** Carbon dioxide ($CO_2$) is a greenhouse gas that plays a crucial role in the Earth's climate system. It is naturally present in the atmosphere, but human activities such as burning fossil fuels, deforestation, and industrial processes have led to a significant increase in $CO_2$ concentrations. When $CO_2$ is emitted into the atmosphere, it absorbs some of the infrared radiation (heat) that is emitted by the Earth's surface and re-emits it back towards the Earth. This process is known as the greenhouse effect.

## Example-5

**Prompt:** Classify the following incident as a breach of protocol. Output 1 for breach, and 0 for no breach: Using a school laptop for personal use.

---

**ARGS:** Taking a school laptop off the school premises. Taking a school laptop off the school premises without permission. Taking a school laptop off the school premises without permission and using it for personal use. Taking a school laptop off the school premises without permission and using it for personal use and downloading personal files onto the laptop. Taking a school laptop off the school premises without permission and using it for personal use and downloading personal files onto the laptop and deleting school files from the laptop.

---

**CD⁻⁻:** Using a school laptop to access non-school websites. Using a school laptop to access social media sites. Using a school laptop to access email accounts not associated with the school. Using a school laptop to access personal files. Using a school laptop to access personal files stored on a USB drive. Using a school laptop to access personal files stored on a cloud service. Using a school laptop to access personal files stored on a personal computer. Using a school laptop to access personal files stored on a personal mobile device. Using a school laptop to access personal files stored on a personal tablet.

---

**TQ⋆ (Ours):** A student using a school laptop for personal use is a breach of protocol because it violates the school's policy on acceptable use of technology. The school laptop is provided for educational purposes, and using it for personal use can compromise the security and integrity of the device, as well as consume valuable resources that could be used for educational purposes. This can also result in potential legal and financial consequences for both the student and the school, as well as damage to the school's reputation. Therefore, using a school laptop for personal use is a breach.

