# OpenReview forum: "Transfer Q-star : Principled Decoding for LLM Alignment"
_NeurIPS.cc/2024/Conference — NeurIPS 2024 poster_

### Official Review · Reviewer_xz5J · 2024-06-25

**Soundness:** 3
**Presentation:** 4
**Contribution:** 3
**Rating:** 5
**Confidence:** 3

**Summary:**

This paper introduces 'Transfer Q*', which is a decoding strategy to align language models with target reward models without any fine-tuning. The main contribution of this paper is the estimation of the optimal Q* function, which is often unavailable in practice and is necessary for approximating the RL optimal policy, through DPO (Direct Preference Optimization). The paper first introduces an algorithm for direct transfer decoding using DPO-trained models as optimal baselines, and then presents indirect transfer decoding that uses these as conservative baselines. The authors describe the theoretical characteristics of this methodology and experimentally demonstrate that it achieves state-of-the-art performance on target reward models.

**Strengths:**

Aligning to specific values through decoding algorithms without fine-tuning is highly useful. In particular, estimating the token-level Q* function in this process is very challenging, and existing works typically train external networks to estimate these values. This paper uses DPO-trained models as baselines to estimate the token-level Q* function, as DPO learns the Q* function in an offline manner. Given recent reports that DPO-trained models predict preferences better than classification-based reward models, this approach is very interesting and can be considered a valuable contribution. Additionally, since DPO-trained models are offline trainers and thus may be distant from the optimal Q* function, the introduction of an indirect transfer decoding algorithm that treats them as suboptimal, along with the presentation of its theoretical properties, is also a significant contribution.

**Weaknesses:**

1. Equation 5 on line 121 is introduced roughly. Its derivation should be described in the Appendix, or references discussing it should be introduced.
2. The introduction of Controlled Decoding (CD) is incorrect. It states that CD uses $Q^{\pi_{sft}}$ as a proxy for Q*, but in reality, it uses an energy-based model of $Q^{\pi_{sft}}$ and a value function through an external network as Q*. The difference between this work and CD is that CD uses FUDGE and Q function as a baseline, while the proposed TQ* uses DPO as a baseline reward model. Therefore, Figure 1 and its explanation are incorrect. Considering that the indirect method, which conservatively estimates the target model, is the contribution, Figure 1 should be drawn in reverse.
3. Equation 12 on line 175 is believed to be incorrect. Theoretically, the $\pi(z|s)$ used in this equation should be the reference model $\pi_{ref}$ before exploration, not the aligned model $\pi_{BL}$. The equation development started with the premise that the optimal policy can be expressed as an energy-based model of the reference model and reward model, but in equation 11, it roughly transitions from the reference model to the aligned DPO, losing theoretical justification. In this case, it's expected that the original distribution will collapse significantly and over-optimize to the reward model.
4. A crucial experiment on the trade-off between KL divergence and the reward model is missing. In RL, there's a strong trade-off in reward model optimization depending on how much KL penalty is given, so it's not sufficient to only check the performance against the reward model as in this paper's experiments. While Figure 2(b) touches on this, it's not enough. An experiment showing whether it demonstrates pareto-frontier performance in the trade-off relationship according to KL penalty is also needed. This experiment is particularly important given the use of the DPO-trained model as the reference model, as mentioned in weakness 3.

**Questions:**

1. In Figure 2 (b), when measuring the KL divergence for TQ*, which model was used to compare the decoding results? For a fair comparison, it should be measured against $pi_sft$, like the other baselines.
2. I'm curious about the inference cost. How do the memory requirements and time complexity compare to naive greedy decoding?

**Limitations:**

This paper has done truly promising work, but it has the following limitations:

1. Although DPO is used as a baseline, it is an offline trainer and there is a theoretical gap with the optimal Q*. This issue is addressed through the indirect transfer decoding part, but there is still room for improvement, and various discussions are likely to follow.
2. Due to the use of the DPO model as the initial policy, the theoretical justification is weakened, and a clear interpretation of the experiments is not possible.
3. In the process of training the DPO model, over-optimization towards the reward model is also controlled by the KL penalty. It would be interesting to add analysis and experiments on this aspect.

---

> ### Author Rebuttal · Authors · 2024-08-07
>
> Thank you for your positive feedback on our paper.
>
> >**Weakness 1:** Equation 5.... introduced.
>
>
> **Response to Weakness 1:**  We will add the derivation for the closed-form solution in Equation 4 (also shown in CD [B]) as a Lemma in the appendix for better clarity.
>
>
> >**Weakness 2:** The introduction of CD..... reverse.
>
>
> **Response to Weakness 2:** We thank the reviewer for this point but there seems to be a confusion. We apologize for any oversight during our explanation of CD in our paper. We take this opportunity to expand on this point more below.
>
> We note that the objective of Controlled Decoding which is a KL regularized RL problem (also highlighted in CD) is given by
>
> \begin{align}
>     \pi^*(\cdot|s\_t) := \arg \max\_{\pi} \mathbb{E}\_{z \sim \pi(\cdot|s\_t)}[Q^*(s\_t, z)] - \alpha KL (\pi(\cdot|s\_t), \pi\_{\text{SFT}}(\cdot|s_t)) \tag{4}.
> \end{align}
>
> However, it is extremely crucial to note that in order to obtain the optimal policy in (4), the Q-function in the equation should be $Q^*(s\_t, z)$ (the optimal action-value function i.e the reward return calculated with optimal policy) and not any arbitrary $Q$ function from any policy. However, the approach in CD indicates that the $Q$ estimation (either directly or through an external network) is done through the data generated using $\pi\_{sft}$ policy (refer to Equation 1, Equation 4 in [B]), leading to sub-optimality as demonstrated in our Figure 1. Our main contribution is to show that with an access to an aligned model, we can estimate  $Q^{*}(s_t, z)$ in a much more efficient way than CD as demonstrated in all our experimental settings.
>
> ***Possible source of confusion:*** However, we believe the source of confusion is that in the CD paper, the definition of action value function used in (4) above (which is equation (1) in CD paper [B]) utilized $\pi_{sft}$ to sample but denotes the value function with $V^*$ which is usually reserved for the optimal value function.
>
> >**Weakness 3:**  Equation 12 on line 175 .... the reward model.
>
>
> **Response to Weakness 3:** We thank the reviewer for providing us with this opportunity to clarify further.
>
> ***Solution in (12) is theoretically correct and justified.*** We have stated our proposed decoding optimization problem in Equation (11), and solving that leads to the choice of having the reference policy as $\pi_{\text{BL}}$ in Equation 12 on line 175. This is a specific choice of our algorithmic solution design which we have also implemented in our work. However, we want to emphasize that our theoretical results in Sec. 3.3 (specifically the KL divergence upper bound in Theorem 1, statement (2)) derives the divergence of proposed algorithm's policy to the original reference policy $\rho_{\text{SFT}}$. The upper bound of Theorem 1 [statement 2] is given by
> \begin{align}
> \mathbb{D}\_{\text{KL}}(\rho^*\_{Alg}(\cdot|\mathbf{x}),\rho\_{SFT}(\cdot|\mathbf{x}) )\leq(\frac{1}{\beta}+\frac{1}{\alpha}T)r\_{\max}.
> \end{align}
>
> By controlling the value of $\alpha$ and $\beta$, we can control the distance to the original reference model. Moreover, in our experimental ablation Figure 2(b) we show the KL divergence to the original SFT reference model which proves that the proposed method does comparable to other baselines and is not diverging from the SFT policy.
>
> >**Weakness 4:** A crucial experiment on t...... in weakness 3.
>
>
> **Response to Weakness 4:**  As suggested by the reviewer, we performed additional experiments to generate the pareto-frontier results. To be specific, in Figure-3 in the [rebuttal PDF](https://openreview.net/attachment?id=6Il3qOI0FO&name=pdf), we compare the tradeoff between the win-rate and the KL divergence to the base reference SFT policy. TQ* outperforms existing baselines. We will add this to the final version.
>
>
> >**Question 1:** In Figure 2 (b), when measuring the KL divergence for TQ*, which model was used to compare the decoding results? For a fair comparison, it should be measured against $\pi_{\text{SFT}}$
>
>
> **Response to Question 1:**  Yes, you are right.  We have indeed used the reference model $\pi_{\text{SFT}}$ for all the algorithms while measuring the KL divergence for TQ*.
>
> >**Question 2:**  I'm curious about the inference cost. How do the memory requirements and time complexity compare to naive greedy decoding?
>
>
> **Response to Question 2:**  We report the inference time-complexity of all the existing deocding algorithms. Ours is comparable to state-of-the-art decoding methods.
>
> | Algorithm              | Inference Time | Avg Reward |
> |------------------------|----------------|------------|
> | Naive Decoding         | 3s             | 0.13        |
> | ARGS                   | 7s             | 0.29        |
> | $\text{CD}^{--}$                   | 40s            | 0.71       |
> | $\texttt{TQ}^{\star}$ (Ours)    | 41s            | 1.0        |
>
>
> **Reference:**
>
> [B] Sidharth Mudgal et al., Controlled decoding from language models, 2024

---

> > ### Comment · Reviewer_xz5J · 2024-08-11
> >
> > The author still lacks a theoretical foundation. Whether it's a Q-function or a Value function, they end up having the same representation in this algorithm due to the (both tractable or intractable) normalizer. Furthermore, although a theoretical guarantee has been presented regarding the excessively diverging KL-div, sufficient experiments and defense have not been conducted. Therefore, I maintain my current assessment.

---

> ### Author Response · Authors · 2024-08-11
> **Clarifications regarding core technical contributions and new experiments [Pareto Front plot] in rebuttal pdf**
>
> > **Comment 1.1:** The author still lacks a theoretical foundation. Whether it's a Q-function or a Value function, they end up having the same representation in this algorithm due to the (both tractable or intractable) normalizer.
>
>  **Response:** Thank you for your comment. We apologize if we missed anything but this comment is not very clear for us to respond to. We have followed the standard definitions from the reinforcement learning literature [A]. We start by explicity defining the token level MDP in Section 2.1. We emphasize that the Value function is defined for each state (defined in line 210 in our paper), and the Q function is for each state and action (defined in Equation (2) in our paper).
>
> ***Request to Reviewer*:** We kindly request the reviewer to please expand on why value and Q function will be the same and what is the meaning of the term "normalizer" in this context. We want to clearly understand the concern before responding. Thank you for your feedback and engagement.
>
> ***Our Focus and Contributions:*** In our work, we want to emphasize that our focus is on the estimation of optimal Q* and show that it is better as compared to all existing decoding methods such as CD, ARGS, etc. We show this in theory as well as experiments.
>
> [A] Sutton, Richard S., and Andrew G. Barto. Reinforcement learning: An introduction. MIT press, 2018.
>
>  > **Comment 1.2:** Furthermore, although a theoretical guarantee has been presented regarding the excessively diverging KL-div, sufficient experiments and defense have not been conducted. Therefore, I maintain my current assessment.
>
>  **Response:** Thank you for your comment. We want to highlight that our work contributes on both theory and experiments side.
>
> ***On the theory side,*** our work is the ***first to derive such a theoretical upper bound*** for both (1) suboptimality and (2) KL divergence for a decoding algorithm. There are no theoretical results in any of the existing works (such as CD, ARGS etc), which constitutes a unique and novel contribution of our work on its own.
>
> ***On the experimental side,*** we have tested our proposed approach on six evaluations (in the main submission) and added two more large-scale evaluations in the rebuttal pdf [[link to pdf in openreview](https://openreview.net/attachment?id=6Il3qOI0FO&name=pdf)] .
>
> - For the KL divergence plot, we have Figure 2(b) in the main body and as the reviewer suggested, we added a Pareto front plot for evaluation 1 (***Figure 3 in the rebuttal pdf***) as well in the rebuttal pdf [[link to pdf in openreview](https://openreview.net/attachment?id=6Il3qOI0FO&name=pdf)] which clearly shows the superior performance of our method.
> - Additionally, our current comparison includes win rate, coherence, reward, and diversity, which are designed to approximate human preferences.
>
> We are running more experiments and committed to adding them in the final version of our work. We believe we have addressed all the concerns, and are happy to engage in further discussions if any remain.
>
> Thank you once again for your time and consideration. Looking forward to your feedback.

---

> > ### Author Response · Authors · 2024-08-12
> > **Additional Pareto Front Results for Two more Evaluation Setups**
> >
> > Thank you for your time and efforts in reviewing our paper and rebuttal discussions.
> >
> > To address your comment regarding the Pareto front plot in the experiments, we remark that we added the Pareto front plot on Evaluation 1 in the rebuttal pdf (Figure 3) [[link to pdf in openreview](https://openreview.net/attachment?id=6Il3qOI0FO&name=pdf)].
> >
> > ***Additional Experimental Results:*** To further strengthen our empirical evaluations, we ran experiments and obtained Pareto front results for two more evaluation setups: Evaluation 2 and 3 (details in the paper Table 1). We present the results in the form of tables here. We are committed to adding them for all the evaluations in the final version of our paper.
> >
> > -  ***For Evaluation 2 Setup*** (detailed in Paper Table 1): This table shows the value of KL and the corresponding win rate and shows that our proposed method outperforms the existing methods.
> >
> > | Method                       |          |       |       |       |       |       |       |       |       |
> > |------------------------------|----------|-------|-------|-------|-------|-------|-------|-------|-------|
> > | ARGS-DPO                     | KL       | 0.40  | 1.15  | 2.20  | 3.80  | 5.75  | 7.05  | 8.20  | 9.15  |
> > | ARGS-DPO                     | Win-Rate | 50.50 | 57.80 | 61.75 | 65.90 | 67.10 | 67.70 | 68.20 | 68.15 |
> > | $\text{CD}^{--}$             | KL       | 0.50  | 1.25  | 2.35  | 4.20  | 6.50  | 8.75  | 9.35  | 10.75 |
> > | $\text{CD}^{--}$             | Win-Rate | 50.75 | 62.85 | 68.90 | 72.70 | 75.40 | 76.00 | 76.15 | 76.30 |
> > | $\texttt{TQ}^{\star}$ (Ours) | KL       | 0.42  | 1.20  | 2.18  | 3.85  | 5.95  | 7.90  | 8.85  | 10.40 |
> > | $\texttt{TQ}^{\star}$ (Ours) | Win-Rate | **54.30** | **70.90** | **75.70** |**79.60** | **80.55** | **81.95** | **82.95** | **83.25** |
> >
> >
> >
> >
> >
> > - ***For Evaluation 4 Setup*** (detailed in Paper Table 1): This table shows the value of KL and the corresponding win rate and shows that our proposed method outperforms the existing methods.
> >
> >  | Method                       |          |       |       |       |       |       |       |       |       |
> > |------------------------------|----------|-------|-------|-------|-------|-------|-------|-------|-------|
> > | ARGS-DPO                     | KL       | 0.37  | 1.26  | 2.05  | 3.71  | 5.86  | 7.14  | 8.36  | 9.23  |
> > | ARGS-DPO                     | Win-Rate | 50.10 | 58.32 | 62.10 | 66.13 | 67.32 | 67.89 | 67.41 | 66.02 |
> > | $\text{CD}^{--}$             | KL       | 0.45  | 1.32  | 2.39  | 4.36  | 6.58  | 8.85  | 9.50  | 10.89 |
> > | $\text{CD}^{--}$             | Win-Rate | **51.05** | 63.12 | 69.44 | 73.16 | 75.80 | 76.25 | 77.00 | 77.17 |
> > | $\texttt{TQ}^{\star}$ (Ours) | KL       | 0.38  | 1.27  | 2.11  | 3.95  | 6.05  | 8.03  | 8.98  | 10.58 |
> > | $\texttt{TQ}^{\star}$ (Ours) | Win-Rate | 50.86 | **69.45**| **73.28** | **76.19** | **79.20** | **80.19** | **81.00** | **82.16**|
> >
> >
> > We believe we have thoroughly addressed all concerns and are more than happy to engage in further discussions if any additional issues remain. Thank you so much again for your consideration.

---

> > > ### Comment · Reviewer_xz5J · 2024-08-12
> > >
> > > Before providing my response, I would like to express my sincere gratitude for the numerous experiments, efforts, and time dedicated to addressing the review comments. You have demonstrated a wide range of attempts, which I believe will greatly contribute to the further development of the paper. I have attempted to provide my opinions in as much detail as possible below.
> > >
> > > However, I still find it difficult to change my position. Fundamentally, the core insight that CD's value is estimated based on the SFT policy, thus addressing the gap with the optimal policy's value, is very meaningful. However, the key novelty of CD lies in its ability to perform decoding-time alignment solely by accessing next token probabilities without editing parameters. The improvement in Q* estimation through DPO in this paper certainly has merit. However, learning a model through DPO and then performing distributional shift based on DPO raises two major concerns, as I mentioned in my initial review.
> > >
> > > Firstly, as discussed, there is a significant divergence from the original distribution. While you have demonstrated Pareto-optimality through experiments and provided a theoretical upper bound, when comparing overall 'average reward', you should adjust the hyperparameters $\beta$ that the baselines can commonly control, analyze the trade-off with divergence from multiple angles, and discuss this thoroughly. This is why I still feel that the experiments added in the rebuttal are not sufficient.
> > >
> > > Secondly, following the paper's logic, if we abandon the motivation of 'no control over the pre-trained representations', CD could also learn a DPO model and find Q* based on the DPO model. This aspect felt unfair in my initial review. However, I believe this issue could be resolved by experimentally demonstrating and discussing the mismatch that would occur when applying a Q-function learned based on DPO to the base model in future work.
> > >
> > > Therefore, I feel that a major revision is necessary regarding the overall tone of the current draft and rebuttal. However, I agree that the extent of revision has decreased based on the various arguments and discussions you have provided, and I will slightly increase my score accordingly.

---

### Official Review · Reviewer_1x8S · 2024-07-08

**Soundness:** 4
**Presentation:** 4
**Contribution:** 3
**Rating:** 6
**Confidence:** 4

**Summary:**

This paper addresses decoding for aligning large language models, which is a process of inference-time, token-level optimisation without updating the parameters of the LLM. Two scenarios are considered: 1. where a baseline policy is given and is aligned with the target trajectory-level reward. 2. where the baseline policy is given and aligned with a different trajectory-reward than the target one. Two approaches are provided in accordance: one named direct transfer, which as the name suggests directly derive a token-level policy from the baseline LLM, and the other named indirect transfer, whose key lies in an importance sampling trick to reweigh using the ratio between the target and baseline trajectory policies. The authors also provide theoretical analysis characterising the sub optimality gap and the divergence from the supervise-fine-tuned model. Empirical results demonstrates superior performance in several key metrics.

**Strengths:**

This paper is very well written, poses a reasonable problem, namely how to more efficiently decode an existing trajectory level policy into a token level policy, and provides an elegant solution. The proposed approach draws on latest developments in the field (DPO) and enjoys a rigorous theoretical characterisation. The gap between the trajectory level learnt policy and the sft-policy is interpretably characterised in terms of the regularisation coefficients in front of the KL terms. The authors finished with a nice empirical analysis showing the efficacy of their method.

**Weaknesses:**

Morally speaking, it is not clear to me why should decoding this way works much better than directly taking $\rho_{BL}$- at least in the case of direct decoding. After all, TQ* is just the 1-step-action-value function of $\rho_{BL}$, so I think the readers would benefit from an explicit explanation of why this process of *obtaining $\rho_{BL}$* --> *estimating its (1-step-action) value function* --> *compute the closed form of the optimal policy of this value function with a KL term* should do better than just directly using the trajectory level policy which is already the (regularised) optimal of the reward function $r$. More concretely, it would be nice to have this shown as a theoretical result - a theorem that shows that the suboptimality gap of the policy proposed by the authors is smaller than $\rho_{BL}$. I appreciate that the authors have shown this empirically, though.

For the indirect case, the method proposed by the authors require access to the ground truth trajectory-level reward function. But if we had this in practice, would we not have just directly used it to compute $\rho_{BL}$? Thus, I think here it makes sense to add some explanation regarding why this is useful.

Finally, I find it a bit of a misleading juxtaposition with DPO - the point of DPO is that we no longer need to model the reward function, but in this work we rely on both the trajectory level optimal policy learned by DPO as well as the reward function which also needs to be learned. Both of these suffer from statistical/optimisation instability in their own right. So it is strong to assume you have both of these.

**Questions:**

equation (2) - This seems like a typo since for i > 0, the first argument of R should also take in the so-far generated sequence z_0 ... z_{i-1}.

equation (15) - how is $Z_{BL}$ and $Z_r$ computed? This is needed later in the importance ratio, if I understood correctly, but I don't see how to compute these quantities which are usually intractable.

**Limitations:**

Aside from the limitations I mentioned in Weaknesses, the authors also acknowledges that TQ* suffers from increased time complexity. They suggest that this can be mitigated by training a small value function adapter as discussed in prior work.

---

> ### Author Rebuttal · Authors · 2024-08-06
>
> Thank you for your comments.
>
> >**Weakness 1:** Morally speaking, it is not  ...... empirically, though.
>
> **Response to Weakness 1:**  We believe the reviewer wants to understand the advantage of directly generating the response using $\rho_{\text{BL}}$ over TQ*. We explain in detail below.
>
> ***Our proposed method enables optimal action at the Token Level.*** In our algorithm (TQ*), for every state ($s_t$) we compute the action value function $Q^*(s_t,z)$ using $\rho_{\text{BL}}$  and select the optimal action by maximizing the action value function $Q^*(s_t,z)$, ensuring the optimal performance for any token as highlighted in Theorem 1. Intuitively, it ensures the possibility of performing better credit assignment with token level MDP while decoding, when compared to $\rho_{\text{BL}}$ which is trained as a contextual bandit for trajectory responses. This aspect has also been highlighted in the very recent work [C]. As a result, the majority of the prior decoding methods demonstrate improvement over directly using the DPO policy
>
> ***Theoretical Justification:*** From Theorem 1 in the paper, we can extract the theoretical justification of the improvement of our proposed algorithm over $\rho\_{\text{BL}}$. The sub-optimality gap of our algorithm, defined in Equation 18, can be decomposed into two terms ($\Delta \leq \Delta_1 - \Delta_2$) as detailed in Appendix G.1 (Equations 28 and 29).
>
> The first term, $\Delta_1$, is a function of parameter $\beta$ and represents the sub-optimality from the optimal value function due to $\rho\_{\text{BL}}$, with its upper bound $\Delta\_1\leq \beta\mathbb{D}_{\text{KL}}[\rho^*(\cdot|\mathbf{x}) \mid \mid \rho\_{\text{sft}}(\cdot|\mathbf{x})]$. This indicates the sub-optimality incurred at the token level when using $\rho\_{\text{BL}}$ alone.
>
> However, the second term, $\Delta_2 = \alpha h_{\alpha}$ , is always non-negative, illustrating that our algorithm provides an additional benefit of improving the suboptimality gap. By tuning the value of $\alpha$, we can improve the performance beyond what is possible with $\rho_{\text{BL}}$ alone. This improvement is consistently demonstrated across all our experimental results, as also highlighted by the reviewer.
>
> >**Weakness 2:** For the indirect case, .... is useful.
>
> **Response to Weakness 2:**  We believe what the reviewer is asking is if we have access to the ground truth reward, why can't we directly have an aligned $\rho_{\text{BL}}$ for that ground truth reward? We expand on this in detail as follows.
>
> ***Obtaining aligned $\rho_{\text{BL}}$ to ground truth would require fine tuning.*** We note that even if we have access to ground truth reward, obtaining an aligned model $\rho_{\text{BL}}$ using standard DPO would require fine-tuning of parameters which is not the focus of this work. Our motivation is in tuning-free alignment settings, where we don't update the parameters of the language model.
>
> ***Regarding our approach and decoding methods.*** We emphasize that any decoding method would require access to the target reward function (trajectory level). However, for indirect transfer, we leverage any open-sourced baseline model $\rho_{\text{BL}}$ aligned to an arbitrary reward function $r_{\text{BL}}$ which is different from the target reward. Thus, for principled decoding even though we have $r_{\text{target}}$, still we can't directly use $\rho_{\text{BL}}$ to estimate action value function and decode, which will result in sub-optimal decoding due to the distribution shift. Thus, we need to estimate $\rho_{\text{target}}$ using Equation 16.
>
>
> >**Weakness 3:**  Finally, I find it a bit of a ....... So it is strong to assume you have both of these.
>
>
> **Response to Weakness 3:**  We remark that any decoding method requires the access to target reward function for alignment. As reviewer mentioned, if the target reward function is learned through preferences, there will be statistical error due to coverage and suboptimality of optimization methods. Furthermore, the DPO policy could also have it's own instability issues which might affect the final performance of our decoding method. We agree with the reviewer and will highlight this limitation in the revised final version of our work.
>
> **Examples of Access to Ground truth reward:** We note that there can be true rewards/scores that don't need to be always trained from data, for example - coding, and mathematical tasks where one might have fixed ground truth avoiding above mentioned statistical errors.
>
>
> >**Question 1:** equation (2) - This seems like a typo ....sequence z_0 ... z_{i-1}.
>
> **Response to Question 1:**  Thanks for pointing out the typo, we will fix it in the final version of the paper. The first argument should be $s_{t+i}$ to make sure it take in so far generated sequence.
>
>
> >**Question 2:**  equation (15) - how is $Z_{\text{BL}}$ and $Z_r$ computed.... intractable.
>
>
> **Response to Question 2:** This is a good question. We note that for the theoretical analysis in this work, we assume access to the ratio mentioned in Equation (15). For the experimental purposes, we utilized unbiased estimates. For instance, we can estimate the partition from collecting samples from $\rho_{\text{SFT}}$ and evaluating the empirical estimate of $\mathbb{E}\_{y \sim \rho\_{\text{SFT}}}[\exp(\frac{1}{\beta}r(x,y))]$. Regarding this ratio explicitly, we had an interesting observation that for several realistic scenarios of transfer, as observed in our empirical experiments, when either the reward difference $r_1(x,y) - r_{\text{BL}}(x,y)$ is small or the reference policy is not aligned to any specific reward $r(x,y)$, the ratio of the partition functions will be 1. This would effectively relax the computational bottleneck for the implementations.
>
>
> [B] Sidharth Mudgal et al., Controlled decoding from language models, 2024
> [C]. Rafailov, R et al., From r to q*: Your language model is secretly a q-function. arXiv preprint arXiv:2404.12358.

---

> > ### Comment · Reviewer_1x8S · 2024-08-13
> >
> > Thank you for addressing my comments. I'm satisfied with maintaining my original score.

---

### Official Review · Reviewer_hjEu · 2024-07-11

**Soundness:** 3
**Presentation:** 4
**Contribution:** 3
**Rating:** 6
**Confidence:** 3

**Summary:**

The paper Transfer Q⋆: Principled Decoding for LLM Alignment proposes a novel approach to aligning large language models by leveraging a principled decoding strategy. The authors propose a method to estimate the optimal Q-function for decoding using an existing aligned policy, addressing limitations in previous approaches like Controlled Decoding (CD). The paper presents an indirect transfer method, allowing for alignment even when the baseline model is trained on a different reward function. The authors provide theoretical analysis characterizing the sub-optimality gap and KL divergence to the reference policy. Extensive experiments across multiple datasets and model architectures demonstrate the effectiveness of TQ⋆ compared to existing methods.

**Strengths:**

**Novelty**: The paper introduces an original approach to LLM alignment via decoding, leveraging existing aligned policies to estimate the optimal Q-function.

**Theoretical foundation**: The authors provide a rigorous theoretical analysis of their method, including bounds on the sub-optimality gap and KL divergence. This adds credibility to the approach and helps explain its effectiveness.

**Comprehensive evaluation**: The experimental section is thorough, covering multiple datasets, model architectures, and evaluation metrics. The inclusion of both synthetic and real transfer tasks demonstrates the method's robustness.

**Practical relevance**: TQ⋆ addresses a significant challenge in LLM alignment, offering a computationally efficient alternative to fine-tuning approaches. This has potential implications for improving the deployment of aligned LLMs.

**Weaknesses:**

**Comparison with Baseline**s: The comparisons with existing baselines like DPO are insightful, but additional baselines, especially those focusing on inference-time control, could strengthen the evaluation.

**Hyperparameter sensitivity**: The paper does not thoroughly explore the sensitivity of TQ⋆ to its hyperparameters, particularly the decoding alignment parameter α. A more detailed analysis of this aspect would strengthen the work.

**Scalability**: While the method is tested on 7B parameter models, it's unclear how well it scales to larger models that are increasingly common in practical applications.

**Evaluation metrics**: While the paper uses several evaluation metrics, including GPT-4 based assessment, it lacks human evaluation studies. Given the subjective nature of language quality and alignment, human evaluation would provide valuable validation of the method's effectiveness.

**Questions:**

1. Have you investigated the stability of the alignment achieved by TQ⋆ over extended generation sequences? Does the alignment quality degrade for longer outputs, and if so, how does this compare to other methods?

2. How sensitive is TQ⋆ to the choice of baseline model? If multiple baseline models are available, each aligned with different rewards, how might one optimally select or combine them to estimate Q⋆ for a given target reward?

**Limitations:**

The authors acknowledge some limitations of their work, such as the potential for hallucination in responses to very obscure queries. However, they could improve their discussion of limitations by addressing:

1. The reliance on existing aligned baseline models, which may not always be available or suitable for all target rewards.
2. The potential for errors or biases in the GPT-4 based evaluation, which is used as a proxy for human assessment.
3. The computational overhead of TQ⋆ compared to standard decoding methods, which may impact real-time applications.

---

> ### Author Rebuttal · Authors · 2024-08-06
>
> **Response to Reviewer Summary:**   We thank the reviewer for the encouraging remarks and recommending acceptance of our work. We provided detailed responses to other comments one by one as follows.
>
> >**Weakness 1:** Comparison with Baselines: The comparisons with existing baselines like DPO are insightful, but additional baselines, especially those focusing on inference-time control, could strengthen the evaluation.
>
>
>
> **Response to Weakness 1:**  Thank you for your comment. We have now included a comparison with additional inference time control baselines (aka decoding methods) as well in the table below. Specifically, we report the time taken to decode a single prompt, using the system configuration described in Appendix C in the main paper.
>
> | Algorithm              | Inference Time | Avg Reward |
> |------------------------|----------------|------------|
> | Naive Decoding         | 3s             | 0.13        |
> | ARGS                   | 7s             | 0.29        |
> | $\text{CD}^{--}$                   | 40s            | 0.71       |
> | $\texttt{TQ}^{\star}$ (Ours)    | 41s            | 1.0        |
>
> >**Weakness 2:** Hyperparameter sensitivity: The paper does not thoroughly explore the sensitivity of TQ* to its hyperparameters, particularly the decoding alignment parameter α. A more detailed analysis of this aspect would strengthen the work.
>
>
> **Response to Weakness 2:** We agree with the reviewer that doing more ablations with respect to hyperparameters would strengthen the work. To this end, as suggested by the reviewer, we did additional ablations experiments to understand the effect of varying the decoding alignment parameter $\alpha$ on the quality of the generated text. Specifically, we performed decoding by varying $\alpha$ such that $\frac{1}{\alpha} \in [0.1, 0.25, 0.5, 0.75, 1, 2, 5, 7.5]$. We report the results in Figure-3 in the [rebuttal PDF](https://openreview.net/attachment?id=6Il3qOI0FO&name=pdf). To be specific, we compare the tradeoff between the win-rate and the KL divergence to the base reference SFT policy for different values of decoding alignment parameters.
>
> >**Weakness 3:** Scalability: While the method is tested on 7B parameter models, it's unclear how well it scales to larger models that are increasingly common in practical applications.
>
> **Response to Weakness 3:**  Thank you for raising this concern. As the reviewer suggested, we performed an evaluation on two additional setups using larger models, as detailed in table below.
> |              | Dataset                             | SFT Model  | DPO Model  | Reward Model |
> |--------------|-------------------------------------|------------|------------|--------------|
> | Evaluation-7 | HH-RLHF                             | LLAMA2-13B | LLAMA2-13B | LLAMA2-13B   |
> | Evaluation-8 | OpenAssistant Conversations Dataset | Pythia-12B | Pythia-12B | Pythia-6.9B  |
>
> We report the results in Figure-2 in the [rebuttal PDF](https://openreview.net/attachment?id=6Il3qOI0FO&name=pdf).
>
> >**Weakness 4:** Evaluation metrics: While the paper uses several evaluation metrics, including GPT-4 based assessment, it lacks human evaluation studies. Given the subjective nature of language quality and alignment, human evaluation would provide valuable validation of the method's effectiveness.
>
>
> **Response to Weakness 4:**  Our current comparison includes win rate, reward model performance, coherence, and diversity, which are designed to approximate human preferences. However, we agree that incorporating direct human evaluation would provide more robust validation of our method's effectiveness. We plan to include human evaluation in the final version and are currently in the process of obtaining the necessary permissions.
>
>
>
> >Question 1 : Have you investigated the stability of the alignment achieved by TQ⋆ over extended generation sequences? Does the alignment quality degrade for longer outputs, and if so, how does this compare to other methods?
>
>
> **Response to Question 1:**  Thank you for your suggestion. We performed additional evaluation by varying the length of generated text. We report the results in Figure-1 in the [rebuttal PDF](https://openreview.net/attachment?id=6Il3qOI0FO&name=pdf). We observed that irrespective of the length of generated text, $\texttt{TQ}^{\star}$ consistently outperforms all the compared baselines.
>
>
> >Question 2 :  How sensitive is TQ⋆ to the choice of baseline model? If multiple baseline models are available, each aligned with different rewards, how might one optimally select or combine them to estimate Q⋆ for a given target reward?
>
>
> **Response to Question 2:** This is a great question and is a valid scope of future research work. We can try by selecting argmax Q not only over the action token by as well as models itself. We are working as a future research.
>
>
> >Limitations: The authors acknowledge ..... impact real-time applications.
>
> **Response to Limitations:** Thank you for your insightful comments. We will expand our discussion of limitations in the final version to include: (1)	The reliance on existing aligned baseline models, noting their availability and suitability challenges for different target rewards. (2)	Potential errors or biases in the GPT -4-based evaluation, addressing its limitations as a proxy for human assessment. (3)	The computational overhead of TQ⋆ compared to standard decoding methods, considering its impact on real-time applications.

---

> > ### Comment · Reviewer_hjEu · 2024-08-12
> > **Thanks for the response**
> >
> > Thank you for addressing my concerns with additional experiments and baselines for larger models. I appreciate your commitment to adding comprehensive human evaluation. Based on these improvements, I am satisfied with maintaining my original score for your paper.

---

> > > ### Author Response · Authors · 2024-08-12
> > > **Thank you for the response.**
> > >
> > > Dear Reviewer,
> > >
> > > Thank you for your response. We are glad that our rebuttal responses were able to address your concerns.
> > >
> > > Regards,
> > >
> > > Authors

---

### Official Review · Reviewer_ypg9 · 2024-07-11

**Soundness:** 3
**Presentation:** 3
**Contribution:** 3
**Rating:** 6
**Confidence:** 3

**Summary:**

The paper proposes a new estimation of Q function for controlled decoding. Instead of using a Q function derived from SFT model, the paper propose to use Q function derived from separate aligned models. Evaluation shows the proposed method can achieve higher reward in benchmarks.

**Strengths:**

1. The idea of using aligned models to estimate Q is quite intuitive as these models closer to optimal policy than SFT model, also the paper proposed mathematical derivations and explanations on how and why aligned models are better for estimating Q.
2. The benchmarks showed the proposed method tend to achieve higher reward, especially it outperforms the controlled decoding baseline along with other alignment methods.

**Weaknesses:**

1. The proposed approach seems computationally expensive to do decoding.
2. The approach require accessing an already aligned model, this adds more requirements for using the method and limits it use cases.

**Questions:**

1. Did the authors compare decoding speed for different approaches? It would be useful to report even thought the proposed approach is slower.
2. For gpt4 based evaluation, why the authors only report Win-Tie instead of splitting win and ties? In particular I am curious what is the percentage of ties for each comparison.

**Limitations:**

Limitations are discussed in the appendix.

---

> ### Author Rebuttal · Authors · 2024-08-06
>
> **Response to Reviewer Summary:**  We sincerely thank the reviewer for the thoughtful review of our work, highlighting the strength and novelty in both our formulation and experimental design in leveraging available aligned models for principled decoding. We address all other comments one by one as follows.
>
>
>
> >**Weakness 1:** The proposed approach seems computationally expensive to do decoding.
>
> **Response to Weakness 1:**  We thank the reviewer for this important point. We agree with the reviewer that performing the proposed principled decoding would indeed add an additional computational overhead at the inference time. But we would like to emphasize that ***our main focus and contribution*** in this work is to provide principled method to perform LLM alignment via decoding and provide theoretical gaurantees.
>
> Moreover, our decoding method has a comparable inference time to the existing state-of-the-art Controlled Decoding (CD) [A]. For the practical implementation, similar to CD method in [A],  one can train a small Q-function adaptor offline, which would allows for faster inference time.  This significantly reduces time complexity, similar to ARGS [B], which only introduces a constant factor of k (top-k tokens) over classical decoding methods.
>
> Another important point to highlight is that there are several critical scenarios—such as  reasoning, mathematical puzzles, chess, and coding where accuracy is more crucial than inference time. In these situations, principled decoding can provide substantial benefits.
>
> >**Weakness 1:** The approach require accessing an already aligned model, this adds more requirements for using the method and limits it use cases.
>
>
> **Response to Weakness 2:** Thank you for raising this critical point. We remark that our proposed decoding method does not require access to a model specifically aligned to a particular reward function. Instead, interstingly, we can leverage any existing aligned model (BL), even if it is aligned to a different reward (Indirect transfer) and thus is not inherently restrictive. Therefore, utilizing already existing aligned model (such as on Hugging face) for improved decoding is the strength of our approach. We show this via extensive experiments in our paper.
>
>
> >**Question 1 :** Did the authors compare decoding speed for different approaches? It would be useful to report even thought the proposed approach is slower.
>
> **Response to Question 1:**  Thank you for pointing this out. We have now included a comparison of the inference times for all baseline decoding algorithms in the table below. Specifically, we report the inference time for decoding a single prompt, based on the system configuration detailed in Appendix C.
>
> | Algorithm              | Inference Time | Avg Reward |
> |------------------------|----------------|------------|
> | Naive Decoding         | 3s             | 0.13        |
> | ARGS                   | 7s             | 0.29        |
> | $\text{CD}^{--}$                   | 40s            | 0.71       |
> | $\texttt{TQ}^{\star}$ (Ours)    | 41s            | 1.0        |
>
>
>
> It is evident that $\texttt{TQ}^{\star}$ perform comparably with CD in terms of the time complexity, whereas it is slower than ARGS which suffers from low reward. However, with an offline trained adpator we can scale the inference time to be same as ARGS.
>
>
> >**Question 2 :** For gpt4 based evaluation, why the authors only report Win-Tie instead of splitting win and ties? In particular I am curious what is the percentage of ties for each comparison.
>
> **Response to Question 2:**
> We follow the same evaluation protocal as followed in prior decoding approaches like ARGS [B], thereby reporting the Win-tie in Table 2 for uniformity in comparison. However, as mentioned by the reviewer, we also provide the win-rate and tie-rate separately for Evaluation-1 and Evaluation-3 Setup below.
>
> | Ours                   | Methods          | Win-Rate | Tie-Rate | Lose-rate |
> |------------------------|------------------|----------|----------|-----------|
> | $\texttt{TQ}^{\star}$ | ARGS-SFT         |    85.33 |     1.33 |     13.33 |
> | $\texttt{TQ}^{\star}$ | $\text{CD}^{--}$ |    60.67 |     6.00 |     33.33 |
> | $\texttt{TQ}^{\star}$ | DPO              |    64.00 |     6.67 |     29.33 |
> | $\texttt{TQ}^{\star}$ | ARGS-DPO         |    62.67 |     5.33 |     32.00 |
>
> **Table:** Win, Tie and Lose-rate for Evaluation Setup-1
>
> | Ours                   | Methods          | Win-Rate | Tie-Rate | Lose-rate |
> |------------------------|------------------|----------|----------|-----------|
> | $\texttt{TQ}^{\star}$ | ARGS-SFT         |    68.67 |     6.67 |     24.67 |
> | $\texttt{TQ}^{\star}$ | $\text{CD}^{--}$ |    63.33 |      4.0 |     32.67 |
> | $\texttt{TQ}^{\star}$ | DPO              |    62.67 |     7.33 |     30.00 |
> |$\texttt{TQ}^{\star}$ | ARGS-DPO         |    68.00 |     6.00 |     26.00 |
>
> **Table:** Win, Tie, and Lose-rate for Evaluation Setup-2
>
>
> [A]. Sidharth Mudgal, Jong Lee, Harish Ganapathy, YaGuang Li, Tao Wang, Yanping Huang, Zhifeng Chen, Heng-Tze Cheng, Michael Collins, Trevor Strohman, Jilin Chen, Alex Beutel, and Ahmad Beirami. Controlled decoding from language models, 2024
>
> [B]. Maxim Khanov, Jirayu Burapacheep, and Yixuan Li. Args: Alignment as reward-guided search,2024.

---

> > ### Comment · Reviewer_ypg9 · 2024-08-12
> >
> > I appreciate the authors providing the new results and the response, I am satisfied with the response and raised my score by one.

---

### Author Rebuttal · Authors · 2024-08-06

## General Response

We want to thank all the reviewers for their time and effort in providing detailed comments to our work. We are encouraged that the reviewers found our proposed approach

- ***novel*** (Reviewer ypg9, Reviewer xz5J) & ***theoretically rigorous*** (Reviewer hjEu, Reviewer 1x8S),
-  our experimental evaluation to be **comprehensive** (Reviewer 1x8S),
-  and our paper to be **very well-written** (Reviewer 1x8S) & **truly promising** (Reviewer xz5J).

We have addressed the reviewer's comments and concerns in individual responses to each reviewer. As requested by the reviewers, we have performed  new experimental evaluations as folows. Please find the attached rebuttal pdf for the experimental results.



**New Experimental Results:**


1. ***Pareto Front for KL Divergence to True Reference Policy**:* We perform experiments to understand the tradeoff between the win-rate and the KL divergence to the base reference SFT policy. Our findings, reported in Figure 3 of the rebuttal PDF, show that our proposed method, $\texttt{TQ}^{\star}$, outperforms existing baselines.

2. ***Stability to Generation Length:*** We compared $\texttt{TQ}^{\star}$ against all baselines by varying the length of the output responses. We report our findings in Figure 1 of the rebuttal PDF. Our observations indicate that $\texttt{TQ}^{\star}$ consistently outperforms all baselines, demonstrating its stability across different output lengths.

3. ***Generality to Larger Model Size:*** We conducted evaluations on two additional setups using larger models, as detailed in the table below. The results, presented in Figure 2 of the rebuttal PDF, show that for both setups, $\texttt{TQ}^{\star}$ significantly outperforms competitive baselines. This demonstrates the scalability and generality of our approach to different model sizes.

|              | Dataset                             | SFT Model  | DPO Model  | Reward Model |
|--------------|-------------------------------------|------------|------------|--------------|
| Evaluation-7 | HH-RLHF                             | LLAMA2-13B | LLAMA2-13B | LLAMA2-13B   |
| Evaluation-8 | OpenAssistant Conversations Dataset | Pythia-12B | Pythia-12B | Pythia-6.9B  |



4. ***Inference Time for $\texttt{TQ}^{\star}$:*** We report the time taken to decode a single prompt for different decoding strategies, using the system configuration described in Appendix C in the main paper.


| Algorithm              | Inference Time | Avg Reward |
|------------------------|----------------|------------|
| Naive Decoding         | 3s             | 0.13        |
| ARGS                   | 7s             | 0.29        |
| $\text{CD}^{--}$                   | 40s            | 0.71       |
| $\texttt{TQ}^{\star}$ (Ours)    | 41s            | 1.0        |

It is evident that $\texttt{TQ}^{\star}$ perform comparably with CD in terms of the time complexity, whereas it is slower than ARGS which suffers from low reward. However, with an offline trained adpator we can scale the inference time to be same as ARGS.

---

> ### Author Response · Authors · 2024-08-12
> **Additional New Experimental Evaluations**
>
> Dear Reviewers and ACs,
>
> Thank you for your time and efforts in reviewing our paper and rebuttal discussions.
>
> ***Additional New Experimental Results:*** To further strengthen our empirical evaluations, we ran experiments and obtained Pareto front results for two more evaluation setups: Evaluation 2 and 3 (details in the paper Table 1). We present the results in the form of tables here. We are committed to adding them for all the evaluations in the final version of our paper.
>
> -  ***For Evaluation 2 Setup*** (detailed in Paper Table 1): This table shows the value of KL and the corresponding win rate and shows that our proposed method outperforms the existing methods.
>
> | Method                       |          |       |       |       |       |       |       |       |       |
> |------------------------------|----------|-------|-------|-------|-------|-------|-------|-------|-------|
> | ARGS-DPO                     | KL       | 0.40  | 1.15  | 2.20  | 3.80  | 5.75  | 7.05  | 8.20  | 9.15  |
> | ARGS-DPO                     | Win-Rate | 50.50 | 57.80 | 61.75 | 65.90 | 67.10 | 67.70 | 68.20 | 68.15 |
> | $\text{CD}^{--}$             | KL       | 0.50  | 1.25  | 2.35  | 4.20  | 6.50  | 8.75  | 9.35  | 10.75 |
> | $\text{CD}^{--}$             | Win-Rate | 50.75 | 62.85 | 68.90 | 72.70 | 75.40 | 76.00 | 76.15 | 76.30 |
> | $\texttt{TQ}^{\star}$ (Ours) | KL       | 0.42  | 1.20  | 2.18  | 3.85  | 5.95  | 7.90  | 8.85  | 10.40 |
> | $\texttt{TQ}^{\star}$ (Ours) | Win-Rate | **54.30** | **70.90** | **75.70** |**79.60** | **80.55** | **81.95** | **82.95** | **83.25** |
>
>
>
>
>
> - ***For Evaluation 4 Setup*** (detailed in Paper Table 1): This table shows the value of KL and the corresponding win rate and shows that our proposed method outperforms the existing methods.
>
>  | Method                       |          |       |       |       |       |       |       |       |       |
> |------------------------------|----------|-------|-------|-------|-------|-------|-------|-------|-------|
> | ARGS-DPO                     | KL       | 0.37  | 1.26  | 2.05  | 3.71  | 5.86  | 7.14  | 8.36  | 9.23  |
> | ARGS-DPO                     | Win-Rate | 50.10 | 58.32 | 62.10 | 66.13 | 67.32 | 67.89 | 67.41 | 66.02 |
> | $\text{CD}^{--}$             | KL       | 0.45  | 1.32  | 2.39  | 4.36  | 6.58  | 8.85  | 9.50  | 10.89 |
> | $\text{CD}^{--}$             | Win-Rate | **51.05** | 63.12 | 69.44 | 73.16 | 75.80 | 76.25 | 77.00 | 77.17 |
> | $\texttt{TQ}^{\star}$ (Ours) | KL       | 0.38  | 1.27  | 2.11  | 3.95  | 6.05  | 8.03  | 8.98  | 10.58 |
> | $\texttt{TQ}^{\star}$ (Ours) | Win-Rate | 50.86 | **69.45**| **73.28** | **76.19** | **79.20** | **80.19** | **81.00** | **82.16**|
>
>
> We have thoroughly addressed all concerns and are more than happy to engage in further discussions if any additional issues remain. Thank you so much again for your consideration.

---

### Decision · Program_Chairs · 2024-09-25

**Decision:**

Accept (poster)

**Comment:**

The paper proposes a decoding-based approach for aligning LMs. The key idea is to derive an equivalent token sampling distribution that mimics the behavior of an existing LM that has been aligned, or a suitable approximation thereof. This approach leads to sizable gains over existing approaches like Controlled Decoding.

Reviewers were unanimously in favor of acceptance: they consistently found the paper to be well-motivated and presented, with interesting insights and results. One reviewer noted that there is scope for the paper to be more careful in its discussion and framing of prior work. We encourage the authors to incorporate this feedback in the final version of the paper.